



# Atmospheric rivers and associated precipitation patterns during the ACLOUD/PASCAL campaigns near Svalbard (May-June 2017): case studies using observations, reanalyses, and a regional climate model

Carolina Viceto[1], Irina V. Gorodetskaya[1], Annette Rinke[2], Marion Maturilli[2], Alfredo Rocha[1], Susanne Crewell[3]

[1]Department of Physics & Centre for Environmental and Marine Studies (CESAM), University of Aveiro, Aveiro, 3810-193, Portugal
[2]Alfred Wegener Institute, Helmholtz Centre for Polar and Marine Research (AWI), Potsdam, 14473, Germany
[3]Institute for Geophysics and Meteorology, University of Cologne, Cologne, 50969, Germany

*Correspondence to:* Carolina Viceto (carolinaviceto@ua.pt)

**Abstract.** Recently, a significant increase in the moisture content has been documented over the Arctic, where both local contributions and poleward moisture transport from lower latitudes can play a role. This study focuses on the anomalous moisture transport events confined to long and narrow corridors, known as atmospheric rivers (ARs) which are expected to have a strong influence on Arctic moisture amounts, precipitation and energy budget. During the two concerted intensive measurement campaigns, Arctic CLoud Observations Using airborne measurements during polar Day (ACLOUD) and the Physical feedbacks of Arctic planetary boundary layer, Sea ice, Cloud and AerosoL (PASCAL), which took place from May 22 to June 28, 2017, at and near Svalbard, three high water vapour transport events were identified as ARs, based on two tracking algorithms: on 30 May, 6 and 9 June. We explore in detail the temporal and spatial evolution of the events identified as ARs and the associated precipitation patterns, using measurements from the AWIPEV research station in Ny-Ålesund, satellite-borne measurements, several reanalysis products (ERA5, ERA-Interim, MERRA-2, CFSv2 and JRA-55) and HIRHAM5 regional climate model. Results show that the tracking algorithms detected the events differently partly due to differences in spatial resolution, ranging from 0.25 to 1.25 degree, in temporal resolution, ranging from 1 hour to 6 hours, and in the criteria used in the tracking algorithms. Despite being consecutive, these events showed different synoptic evolution and precipitation characteristics. The first event extended from western Siberia to Svalbard, causing mixed-phase precipitation and was associated with a retreat of the sea-ice edge. The second event a week later had a similar trajectory and most precipitation occurred as rain, although in some areas mixed-phase precipitation or only snowfall occurred, mainly over the north-eastern Greenland's coast and northeast of Iceland and no differences were noted in the sea-ice edge. The third event showed a different pathway extending from north-eastern Atlantic towards Greenland, and then turning southeastward reaching Svalbard. This last AR caused high precipitation amounts in the east coast of Greenland in the form of rain and snow and showed no





precipitation in Svalbard region. The vertical profiles of specific humidity show layers of enhanced moisture, simultaneously with dry layers during the first two events, which were not captured by all reanalysis datasets, while the model misrepresented the entire vertical profiles. Regarding the wind speed, there was an increase of values with height during the first and last events, while during the second event there were no major changes in the wind speed. The accuracy of the representation of wind speed by the reanalyses and the model depended on the event. This study shows the importance of both the Atlantic and

Siberian pathways of ARs during spring-beginning of summer in the Arctic, AR-associated strong heat and moisture increase as well as precipitation phase transition, and the need of using high spatiotemporal resolution datasets when studying these intense short duration events.

## 1 Introduction

The Arctic is a region of major interest due to its high sensitivity to global warming with significant implications for both the

regional climate and the global climate system (McGuire et al., 2006). Thus, changes in the Arctic might have implications beyond the region, influencing the mid-latitude climate and weather. For instance, changes during the summer, including a weakening of the storm tracks, a meridional shift in jet position and an amplification of quasi-stationary waves, can increase the persistence of summer hot and dry extremes in mid-latitudes (Coumou et al., 2018). On contrary, some studies point to an increase of the probability of severe winter occurrence in the mid-latitudes (e.g. central Eurasia (Mori et al., 2019) and eastern

United States (Cohen et al., 2018)), due to the Arctic warming.

A significant increase in the atmospheric moisture content has been documented over the Arctic in the recent years (Rinke et al., 2019; Screen and Simmonds, 2010). This is partially explained by the reduction of sea-ice cover, which enhances local evaporation (Bintanja and Selten, 2014). However, others argue that the predominant reason is the enhanced poleward moisture flux during the recent decades (Zhang et al., 2013), which is expected to continuously increase in the future (Bengtsson et al.,

2011; Bintanja and Selten, 2014; Kattsov et al., 2007; Skific and Francis, 2013). This might be due to several factors or a combination of them, such as changes in the atmospheric circulation patterns, increased moisture transport intensity, and/or higher evaporation rates in the lower latitude moisture source regions (Gimeno et al., 2015). However, Gimeno et al. (2019) reason in their review that there is no agreement in calculated trends in atmospheric moisture transport to the Arctic.

Extreme poleward moisture transport events towards the Arctic are known as moisture intrusions. Woods et al. (2013)

identified an average of 14 moisture intrusions per season, for boreal winters from 1990 to 2010, with a typical duration of 2 to 4 days, corresponding to 28 % of the total poleward moisture transport across 70° N. These moisture intrusions have a filamentary structure, showing similar features to a phenomena known as atmospheric rivers (ARs) (Baggett et al., 2016).

Our study focuses on the ARs, which are recognized by an anomalous moisture transport confined to long, narrow and transient corridors. ARs are characterized by a filament of high specific humidity, which is fuelled by the transport of moisture from

(sub-)tropical to higher latitudes and/or the moisture convergence along the pre-cold frontal low-level jet of an extratropical cyclone, which is part of the Warm Conveyor Belt (WCB) (Ralph et al., 2004). Extratropical cyclones are low pressure systems





associated with cold, warm and occluded surface fronts. The water vapour arising from the warm sector of the cyclone converges along the cold front, characterized by cool and dry air, which catches up with the warm front. As a result, a narrow band of high water vapour content is formed ahead of the cold front at the base of the WCB, associated with strong low-level
winds.

Multiple studies have analysed the increase in poleward moisture transport into the Arctic region and the associated impacts, including warming (Johansson et al., 2017), decrease in the sea-ice concentration (Lee et al., 2017; Park et al., 2015; Yang and Magnusdottir, 2017), increase in precipitation (Bintanja et al., 2020; Gimeno-Sotelo et al., 2018) and changes in the cloudiness and cloud radiative heating (Johansson et al., 2017). Although ARs are mostly studied for the western coast of North America
and Europe, they have a remarkable importance for the high latitudes. Previous studies showed that ARs have a strong influence on both Arctic and Antarctic mass and energy budget (Gorodetskaya et al., 2014, 2020; Mattingly et al., 2018; Nash et al., 2018; Neff et al., 2014; Wille et al., 2019; Woods et al., 2013; Woods and Caballero, 2016). The majority of the water vapour flux (90 %) at mid-latitudes (50º N) occurs across the Atlantic and Pacific AR pathways, with around 60 % occurring as ARs (Nash et al., 2018; Zhu and Newell, 1998).

In the Arctic latitudes, the enhanced poleward moisture transport is related to an increase in precipitation (Bintanja et al., 2020; Kattsov et al., 2007; Zhang et al., 2013). The precipitation phase (rain and/or snow) might influence the sea-ice. While fresh snow increases surface albedo in spring-summer thus helping to maintain a colder surface and reducing ice melting, it enhances the thermal insulation and reduces ice growth in late autumn-winter. Rainfall strongly decreases the surface albedo enhancing the melting of the snow/ice (Räisänen, 2008). Concluding, the precipitation phase induces different feedback mechanisms, due
to changes in the surface albedo, and consequent adaptation of the surface energy budget (Callaghan et al., 2011). Furthermore, ARs in the Polar Regions also increase the downward longwave radiation (mostly due to the cloud radiative forcing), which increases the surface temperature and can enhance the retreat of sea-ice extent (Hegyi and Taylor, 2018; Komatsu et al., 2018; Wille et al., 2019) and Greenland ice sheet melt (Bennartz et al., 2013; Mattingly et al., 2020; Neff, 2018; Neff et al., 2014).

The AR detection consists on applying tracking algorithms defined by specific criteria, such as minimum areas, with specific
width and length, where the Integrated Water Vapour (IWV) and/or the Integrated Vapour Transport (IVT) reach or exceed specific threshold values. Shields et al. (2018) presented an extensive list of tracking algorithms, with different criteria to identify ARs. The majority of the algorithms are applied on the Western U.S. (e.g. Dettinger, 2013; Gershunov et al., 2017; Rutz et al., 2014). Only few tracking algorithms were developed and applied for the Polar Regions, specifically to Antarctica (Gorodetskaya et al., 2014, 2020; Wille et al., 2019), and Greenland (Mattingly et al., 2018).

Furthermore, Shields et al. (2018) study aimed to understand and quantify the uncertainties of detecting ARs based only on tracking algorithms and amongst them. AR characteristics such as frequency, duration and intensity were analysed in this study, and although it comprises only a period of one month (February 2017), results already point to differences in these characteristics depending on the algorithms formulation. This study was extended by Rutz et al. (2019) for a longer period (January 1980 to June 2017), which highlighted a wide range of frequency, duration and seasonality results amongst the
algorithms, although their meridional distribution through selected coastal transects (North American and European West



Coasts) was similar across algorithms. With the purpose to address the differences and uncertainties resultant of the application of different tracking algorithms, in this study two detection methods – global algorithm by Guan et al. (2018) and the algorithm developed for Antarctica by Gorodetskaya et al. (2014, 2020) – explained later with further detail, were applied.

Here we present a detailed analysis of three ARs identified in May-June 2017 during two coordinated field campaigns along Svalbard: the Arctic CLoud Observations Using airborne measurements during polar Day (ACLOUD) (Ehrlich et al., 2019; Wendisch et al., 2019), and the Physical feedbacks of Arctic planetary boundary layer, Sea ice, Cloud and AerosoL (PASCAL) (Macke and Flores, 2018; Neggers et al., 2019; Wendisch et al., 2019). We explore their temporal and spatial evolution, and the associated precipitation patterns, using several reanalysis products. Reanalysis-based estimates are compared with the ground-based remote sensing and radiosonde measurements at Ny-Ålesund using the intensive observational period during the ACLOUD/PASCAL campaigns, and satellite-borne measurements. Concurrently, state-of-the-art Arctic regional climate model simulations are evaluated. This study assesses the differences between different reanalysis datasets, their agreement with measurements, and the discrepancies between the model and the reanalyses and measurements. Further, we apply these various observational and modelling products for investigating the ARs development and evolution, their role in the poleward moisture transport (reaching and affecting Svalbard and Greenland), and associated precipitation characteristics. Another purpose of this study is to adapt the AR tracking algorithm by Gorodetskaya et al. (2020), developed originally for Antarctica, to the Arctic region, to evaluate how well does it identify ARs and to identify the most suitable reanalysis dataset to analyse this type of events. Building on this detailed case studies analysis, it will be possible to extend this work to longer time periods from the recent past (using reanalyses) and into the future.

## 2 Data

### 2.1 In situ and remote sensing measurements

We used observations from the French (Polar Institute Paul Emile Victor) and German (Alfred Wegener Institute for Polar and Marine Research) Arctic Research Base (AWIPEV), located in Ny-Ålesund (http://www.awipev.eu/), which consist of a suit of near-surface and ground-based remote sensing long-term observations. In this study we used data from radiosondes, the Humidity And Temperature PROfiler (HATPRO) microwave radiometer and the Global Navigation Satellite System (GNSS) ground station.

Radiosondes are regularly launched in Ny-Ålesund once per day since November 1992 (Maturilli and Kayser, 2017). Since April 2017, the regular sounding is done with Vaisala RS41-SGDP sondes. During the period covering the ACLOUD/PASCAL campaigns, additional radiosondes were launched on a 6-hourly basis, providing vertical profiles of temperature, relative humidity, pressure and wind (Maturilli, 2017a, 2017b). From these high resolution atmospheric parameters, it is possible to derive integrated variables for the atmospheric column, such as the IWV and IVT.

HATPRO is a ground-based microwave radiometer capable of measuring brightness temperatures along a vertical column of air. This instrument operates in two different reception bands: 22.235-31.400 GHz (seven channels in the water vapour band,





sensitive to humidity) and 51.26-58.00 GHz (seven channels in the oxygen band, influenced by temperature), with a temporal resolution of 1-2 seconds (Nomokonova et al., 2019, 2020; Rose et al., 2005). Afterwards, the brightness temperatures are

used to retrieve vertical profiles of humidity and IWV. A quality flag that characterizes the instrument and retrieval performance was applied.

The GNSS ground station, installed in Ny-Ålesund, has a 15 minute temporal resolution and retrieves the IWV content along the zenith path (Bevis et al., 1992). This data was obtained from GeoforschungsZentrum Potsdam (GFZ), who runs the EPOS software to process the data in near-real time (Dick et al., 2001; Ge et al., 2006; Gendt et al., 2004).

Satellite remote sensing measurements from the MetOp polar orbiting satellites provide information on the spatial coverage of the AR. The IASI L2 PPFv6 dataset used in this study combines measurements by the Infrared Atmospheric Sounding Interferometer (IASI, Blumstein et al., 2004), and two microwave instruments, i.e. the Advanced Microwave Sounding Unit (AMSU) and the Microwave Humidity Sounder (MHS). Temperature and humidity vertical profiles are retrieved from which IWV is derived.

**2.2 Reanalysis datasets**

Several reanalysis products were used: 1 – the European Centre for Medium-Range Weather Forecasts (ECMWF) Re-Analysis (ERA) Interim (ERA-Interim), 2 – the ERA5 reanalysis, 3 – the Modern-Era Retrospective analysis for Research and Applications, version 2 (MERRA-2), 4 – the Climate Forecast System version 2 (CFSv2), 5 – the Japanese 55-Year Reanalysis (JRA55). A detailed description of the different reanalysis products is presented in Table 1.

Reanalysis data were downloaded for a period covering the ACLOUD/PASCAL campaigns. To detect the ARs, specific humidity, temperature and meridional and zonal components of the wind were acquired from 1000 hPa to 300 hPa. Since the majority of the reanalysis datasets, with exception of MERRA-2, have the first pressure levels below the surface, we applied a procedure similar to Gorodetskaya et al., (2020), that uses the variable surface pressure to exclude these layers. To ensure a full assessment of the events, mean sea level pressure, potential temperature (at 2 PVU), geopotential (at 700 hPa), sea-ice

area fraction, total precipitation and snowfall data were also obtained.

**2.3 Regional climate model**

The detected ARs and related precipitation were compared to the output of the state-of-the-art atmospheric regional climate model HIRHAM5 (Christensen et al., 2007; Sommerfeld et al., 2015), which participated in recent model intercomparisons within Arctic CORDEX (Inoue et al., 2021; Sedlar et al., 2020). Furthermore, HIRHAM5 has been thoroughly evaluated and

applied for a wide range of Arctic climate studies, which include, for example, quantification of the freshwater input in southwest Greenland (Langen et al., 2015), cyclones activity in the Arctic (Akperov et al., 2018), Arctic 2 meter air temperature (Zhou et al., 2019), and clouds and radiation processes over the Arctic Ocean (Inoue et al., 2021; Sedlar et al., 2020).

This model includes the physical parametrizations of the general circulation model ECHAM5 (Roeckner et al., 2003). Relevant for this paper, the stratiform cloud scheme consists of prognostic equations for the vapor, liquid, and ice phase, respectively,





**Table 1.** Description of the reanalysis products used in this study.

| Data name | ERA-Interim | ERA5 | MERRA-2 | CFSv2 | JRA-55 |
|---|---|---|---|---|---|
| Source | European Centre for Medium-Range Weather Forecasts (ECMWF) | ECMWF | National Aeronautics and Space Administration (NASA) | National Centers for Environmental Prediction (NCEP) | Japan Meteorological Agency (JMA) |
| Period | Jan 1979-Aug 2019 | 1979-present | 1980-present | 2011-present | 1958-present |
| Temporal resolution | 6 hours | 1 hour | 3 hours | 6 hours | 6 hours |
| Spatial resolution | 0.5 degree interpolated from the original 0.75 degree | 0.25 degree | 0.5 x 0.625 degree | 0.5 degree | 1.25 degree |
| Vertical resolution | 37 pressure levels 60 model levels | 37 pressure levels 137 model levels | 42 pressure levels 72 model levels | 37 pressure levels 64 model levels | 37 pressure levels 60 model levels |
| Vertical coverage | 1000 to 1 hPa | 1000 to 1 hPa | 1000 to 0.1 hPa | 1000 to 1 hPa | 1000 to 0.1 hPa |
| References | Dee et al. (2011) | Hersbach et al. (2020) | Gelaro et al. (2017) | Saha et al., (2014) | Kobayashi et al. (2015) |

a cloud microphysical scheme (Lohmann and Roeckner, 1996), and a diagnostic relative humidity based cloud cover scheme (Sundqvist et al., 1989). For precipitation, all relevant microphysical processes and conversions are parametrized; we refer for details to Roeckner et al. (2003).

The applied domain comprises the entire Arctic for latitudes higher than approximately 65° N, with a horizontal resolution of 0.25 degree and 40 vertical levels until 10 hPa and 10 vertical levels in the lowest first kilometre. A more detailed description of the model and its parameterizations can be found in the given references.

ERA-Interim was used to initialize and force HIRHAM5. ERA-Interim fields are used as the lower boundary conditions, namely daily sea surface temperature and sea-ice concentration and the 6 hourly lateral boundary forcing for the prognostic

variables (surface pressure, and profiles of air temperature, horizontal wind components, specific humidity, cloud water and ice). A grid point nudging (e.g., Omrani et al., 2012) was applied with a relaxation scale equivalent to a 1 % nudging in all model levels to constrain the large-scale dynamics.





## 3 Methodology

### 3.1 IWV and IVT

IWV and IVT were calculated for the entire duration of the ACLOUD/PASCAL campaigns, between the first near-surface
level (equal or less than 1000 hPa) and 300 hPa. IWV is derived from specific humidity ($q$) based on the following equation:

$$IWV = -\frac{1}{g}\int_{1000\,hPa}^{300\,hPa} q \; dp \qquad (1)$$

Where $g$ is the acceleration due to the gravity. IVT is based on $q$ and horizontal wind ($\vec{V}$), using Eq. (2):

$$\overrightarrow{IVT} = -\frac{1}{g}\int_{1000\,hPa}^{300\,hPa} q\vec{V} \; dp \qquad (2)$$

### 3.2 Detection of atmospheric rivers

Two tracking algorithms were used to identify ARs: Gorodetskaya et al. (2014, 2020) developed and applied for Antarctica
and Guan et al. (2018) global algorithm. Gorodetskaya et al. (2014) determined an AR when IWV (calculated from 900 to 300
hPa) is equal or higher to a minimum threshold value near the Antarctic coast (within 20° W and 90° E longitudinal sector),
and continuous at all latitudes for at least 20° equatorward (length > 2000 km), within a limited width of 30° longitude (~ 1000
km at 70° S increasing equatorward). This zonal mean threshold is based on saturated IWV and on an AR coefficient that
determines the strength of the AR, which is explained in detail by Gorodetskaya et al. (2014). A second version of the algorithm
included some updates, namely the computation of IWV from the first near-surface level with pressure equal or less than 1000
hPa to 300 hPa, and the longitude width of 40 degree in order to include zonally-oriented ARs (Gorodetskaya et al., 2020).
We adapted this formulation for the Arctic, considering the ARs reaching and crossing 70º N (within 50° W and 110° E
longitudinal sector, according to the considered campaign domain), and continuous at all latitudes for at least 2000 km, within
a maximum width of 40º longitude. The axis of an AR is defined as the maximum value of IWV at each latitude. In this study
we explored the sensitivity of the AR identification in the Arctic to both the threshold and various geometric criteria and have
included also the potential AR events (pAR) when IWV is equal or higher to the threshold (as defined in Gorodetskaya et al.,
2020). If the geometrical criteria are also met this event is classified as an AR. This algorithm will be hereafter referred as
Gorodetskaya2020.
The second tracking algorithm, based on IVT, is fully described in Guan and Waliser (2015) (V1.0). In this case, the
identification of an AR is based on several conditions. First, an IVT threshold for each grid cell is calculated, which results
from the combination of a defined percentile and a fixed lower limit value. Since the Polar Regions are characterized by low
values of IVT, mainly due to lower moisture values, the threshold is defined using the 85$^{th}$ percentile and a lower limit value
of 100 kg m$^{-1}$ s$^{-1}$. If the objects exceed this limit, the IVT direction is evaluated, in a way that the coherence in IVT direction,





the object mean meridional IVT and the consistency between object mean IVT direction and overall orientation are checked.
Then, a filter for the length (minimum 2000 km) and length/width ratio of each object (higher than 2) is applied.

In this study, we used a refined version of this tracking algorithm, described in Guan et al. (2018) (V2.0), which includes the application of successively increasing IVT percentile thresholds (from 85th to 95th percentile, by 2.5th percentile). This algorithm will be referred as Guan2018 in the following sections. Only MERRA-2 reanalysis, covering a period from 1980 to 2019, were used to calculate IVT and to detect the ARs. This database was provided by Bin Guan via https://ucla.box.com/ARcatalog.

### 3.3 Air mass trajectories

The back trajectory model HYbrid Single-Particle Lagrangian Integrated Trajectory (HYSPLIT) from the National Oceanic and Atmospheric Administration (NOAA) (Draxler and Hess, 1998) was used in order to track multiple air masses and establish possible moisture sources. This model computes simple air parcel trajectories, complex transport, dispersion, chemical transformation, and deposition simulations (Rolph et al., 2017; Stein et al., 2015), based on gridded meteorological data archives. For this study we used NCEP's Global Data Assimilation System (GDAS) model, with a horizontal resolution of 0.5 degree. The dates when the ARs reached Ny-Ålesund were used to compute an ensemble of 5 days back trajectories with 27 members. The calculation of each member consists in adding an offset to the meteorological data (one grid point and 0.01 sigma units in the vertical).

## 4 Results

### 4.1 AR detection during ACLOUD/PASCAL campaigns

A synoptic overview on ACLOUD/PASCAL has been presented by Knudsen et al. (2018) where already four events with substantial water vapour transport were identified: 30 May, 6 June, 9 June and 13 June (cf their Figure A1). Here we take a closer look making use of high temporal resolution IWV measurements by HATPRO in Ny-Ålesund, and IWV and IVT by ERA-Interim reanalysis during the complete period of ACLOUD/PASCAL campaigns which depicts strong IWV and IVT variability including distinct IWV maxima on these days (Fig.1). After, these events and their possible association with ARs is analysed.

For times of the highest IWV at Ny-Ålesund during each event we investigate the spatial IWV structure for three reanalysis datasets (ERA-Interim, ERA5 and MERRA-2) with different temporal and spatial resolution, HIRHAM5 model and satellite measurements (Fig. 2). To find which events were identified as pARs or ARs, Gorodetskaya2020 tracking algorithm was applied to the reanalysis and model fields. Note that, polar orbiting satellite measurements with limited swath width are not suitable to detect ARs since the application of the tracking algorithms implies using complete gridded data. The ARs detected by Guan2018 database (only applied to MERRA-2 reanalysis), were also included to compare the differences between both

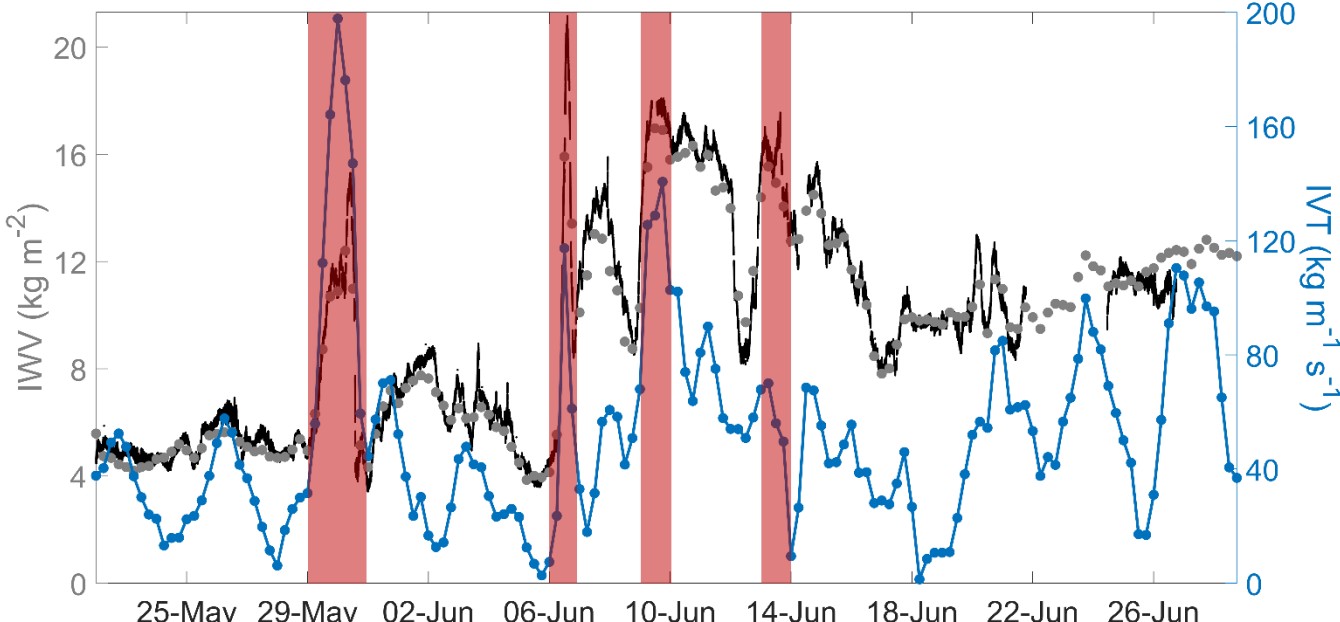


**Figure 1.** Time series of integrated water vapour (IWV, kg m$^{-2}$), based on HATPRO measurements at Ny-Ålesund (black line) and ERA-Interim reanalysis at the closest grid (grey dots), and integrated vapour transport (IVT, kg m$^{-1}$ s$^{-1}$, blue dotted line), based on ERA-Interim reanalysis, for the ACLOUD/PASCAL campaigns (22 May-28 June 2017). Red bars show anomalous IWV and/or IVT at Ny-Ålesund.

tracking algorithms. The area covered by these pARs or ARs and during the 24 hours before and after these times is shown in Figure 3.

After applying Gorodetskaya2020 tracking algorithm, two of the four events were detected as pARs: 30 May and 6 June (Figs. 2a and 2b, red lines; Figs. 3a and 3b, coloured circles). With the inclusion of the geometrical criteria only the first event was identified as an AR (Fig. 2a, magenta line; Fig. 3a, coloured dots). Guan2018 detection algorithm identified two ARs on 30

May and 9 June (Figs. 2a and 2c, white lines; Figs. 3a and 3c, purple squares). The fourth event, on 13 June, was not identified by any tracking algorithm as an AR (and thus is not shown in this paper).

The first event, on 30 May, identified as an AR by both tracking algorithms, was associated to a long and narrow band with high IWV extending westward from Western Siberia (around 60° N, 90° E) to Svalbard archipelago (around 80° N, 15° E) (Fig. 2a). The AR had a similar shape in all reanalysis datasets, although in MERRA-2 and CFSv2 products it extended further

southeast, possibly related to higher values of IWV in these reanalyses over the region (Fig. S1), resulting in a larger area covered by the pAR/AR shapes (Fig. 3a). Focusing only on MERRA-2 reanalysis in order to compare the two algorithms, both show overlapping contours (Fig. 2a). While Gorodetskaya2020 shape was more elongated and extended to lower latitudes, until continental Siberia, covering a larger area (Fig. 3a), Guan2018 shape was confined to the ocean area due to lower values of IVT over land (not shown in the paper).






**Figure 2.** Maps of the integrated water vapour (IWV, kg m$^{-2}$, colour shading) for the times with the highest IWV values in Ny-Ålesund during the 30 May event [first column, **(a)**], 6 June event [second column, **(b)**] and 9 June event [third column, **(c)**] based on reanalyses (ERA-Interim, ERA5 and MERRA-2), HIRHAM5 model and IASI observations. Magenta line shows AR shape (based on Gorodetskaya2020) and red line shows the shape of potential ARs (IWV≥IWVthres, based on Gorodetskaya2020). White line shows AR shape (based on Guan2018) and orange arrows show integrated vapour transport (IVT, kg m$^{-1}$ s$^{-1}$), both based only on MERRA-2 reanalysis. Note that AR shape based on Gorodetskaya2020 might overlap pAR shape in some cases. Black star shows Ny-Ålesund location. Figures S1, S2 and S3 show the complete temporal evolution of the events for all datasets.



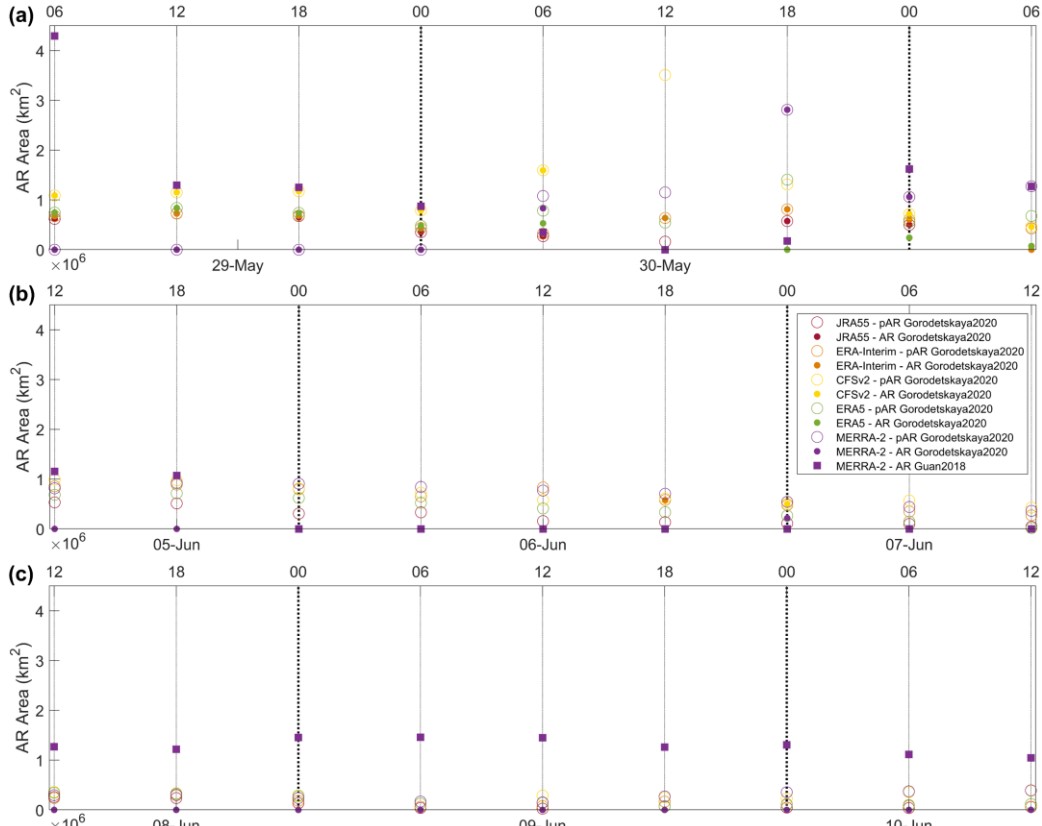

**Figure 3.** Time series of the area of the AR shape (based on Gorodetskaya2020), of the shape of potential ARs (IWV≥IWVthres, based on Gorodetskaya2020), and AR shape (based on Guan2018, only for MERRA-2 reanalysis), during the 30 May event [first row, **(a)**], 6 June event [second row, **(b)**] and 9 June event [third row, **(c)**] based on reanalyses (ERA-Interim, MERRA-2, ERA5, CFSv2, JRA-55).

One week later, on 6 June, the second event identified as a pAR by Gorodetskaya2020, made landfall in Ny-Ålesund (Fig. 2b). This AR resulted from two long and narrow filaments with high IWV also extending from Western Siberia, converging into one wider filament near Novaya Zemlya. The pAR shape was similar in ERA-Interim and ERA5, but for MERRA-2 and CFSv2 it extended further southeast due to the higher values of IWV over continental Siberia compared to ERA-Interim, ERA5 and JRA-55 reanalyses (Fig. S2). No major differences were noticed in the area of the pAR/AR shapes (Fig. 3b). Events like this, with a strong zonal component, are not identified as ARs by both algorithms, due to limitations in the definition of the tracking algorithm, however Gorodetskaya2020 algorithm identifies it as pAR before applying geometrical criteria. Due to the strong zonal component and a complex shape of this pAR, the event was not identified as a full AR by the strict geometrical criteria in Gorodetskaya2020 algorithm. Currently the geometric criteria in Gorodetskaya2020 algorithm are being adapted as such zonal events must be taken into account in future studies, when applying this and other algorithms to long-term analysis. Three days later, on 9 June, the third event was identified by Guan2018 tracking algorithm as an AR, while it did not fulfil the criteria defined by Gorodetskaya2020 algorithm. However, small pARs areas were also identified using the latter algorithm,





but, since there is no consecutive shape inferred, the adaptation of the geometrical criteria used in the algorithm would still not include these areas as a full AR (Figs. 2c and 3c). Note that overall, Guan2018 global algorithm is much less restrictive compared to the polar-specific algorithms (Rutz et al., 2019). This event reached Ny-Ålesund extending north-westward from north-eastern Atlantic (near Scandinavian Peninsula) towards Greenland, passing over the northeastern region of Greenland, and then turning southeastward eventually reaching Svalbard from the north. A similar IWV pattern was found in all

reanalyses.

These bands of high IWV were observed in all reanalysis datasets and HIRHAM5 model, despite some differences in the amount of IWV and in the shape of the pAR/AR. These discrepancies might be related with different spatial and temporal resolutions and data assimilation of the reanalysis products and the model. In general, the comparison of the reanalysis datasets and HIRHAM5 model with IASI measurements shows similar amounts and location of the bands of high moisture content. A

more quantitative assessment of different IWV datasets including further satellite products has been carried out by Crewell et al. (2021).

A complete spatiotemporal evolution of the three events, including the maps for 6 hours previous and after the IWV peaks and all the reanalysis products, is shown in Figs. S1, S2 and S3. Comparing the events, the first two extended from Western Siberia while the last extended from Scandinavia, however, despite these differences, the three events were intense short-duration

events.

In the following sections, we provide a detailed analysis of the three events detected as pARs/ARs.

## 4.2 Synoptic conditions during ARs affecting Svalbard

To understand which meteorological conditions triggered the detected events, their synoptic situation was analysed. For this purpose, we performed a detailed analysis of the synoptic conditions focusing only in the days when the pARs/ARs reached

Ny-Ålesund, using ERA5 reanalysis, due to its high temporal and spatial resolution.

Figure 3 (Figs. S4 and S5 for temporal evolution) shows the mean sea level pressure (MSLP), the geopotential height at 700 hPa and potential temperature (θ) at 2 potential vorticity units (PVU), which is commonly used to define the height of the dynamical tropopause (Hoskins et al., 1985; Juckes, 1994; Wilcox et al., 2012; Woollings et al., 2018) providing an analysis of the upper-level flow. The combination of these variables is usually used to study the atmospheric blocking, which has

previously been associated to ARs (Benedict et al., 2019; Francis et al., 2020; Rabinowitz et al., 2018; Wille et al., 2019). The atmospheric blocking leads to persistent weather conditions, playing an important role in directing ARs poleward. This phenomena has a wide range of consequences, ranging from persistent high/low temperatures to hydrological impacts (Woollings et al., 2018). Knudsen et al. (2018) mentioned that during the warm period of ACLOUD/PASCAL campaigns (from 30 May to 12 June), moderate negative Arctic Oscillation index values were found, which are related to more frequent

blocking high-pressure events.

During the first event a low-pressure system was centred over the Barents Sea, with a blocking high-pressure ridge in the polar latitudes (Fig. 4a). These systems remained almost stationary although the cyclone slightly moved southwestward and





weakened (Fig. S4a). Simultaneously, low potential temperatures were found in the location of the low-pressure system (Fig. S5a), as expected, following the slow cyclone propagation towards southwestward direction (Fig. S4a). In the region of the

AR, relative high values of potential temperatures were noticed, associated to the vertical advection of potential temperature (Fig. S5a). This displacement directed the moisture transport and the associated AR westward from the lower latitudes in Siberian towards higher latitudes around Svalbard, followed by a small shift in direction to southwestward.

One week later, a stronger low-pressure system affected the southern region of the Svalbard archipelago along with a high-pressure system at higher latitudes, less pronounced than the previous event (Fig. 4b). The cyclone progressed northwestward

from northern Scandinavia, and slowly moved towards Greenland with no intensity changes (Fig. S4b). At the same time, a second weaker low-pressure system located in northern Russia caused the tilt of one of the pAR branches to a zonal direction. These almost stationary systems, associated to atmospheric blocking, directed the moisture transport from Western Siberia to southern Svalbard. In the region of the pAR and north of its shape even higher values of potential temperatures were found than in the previous AR (Fig. S5b).


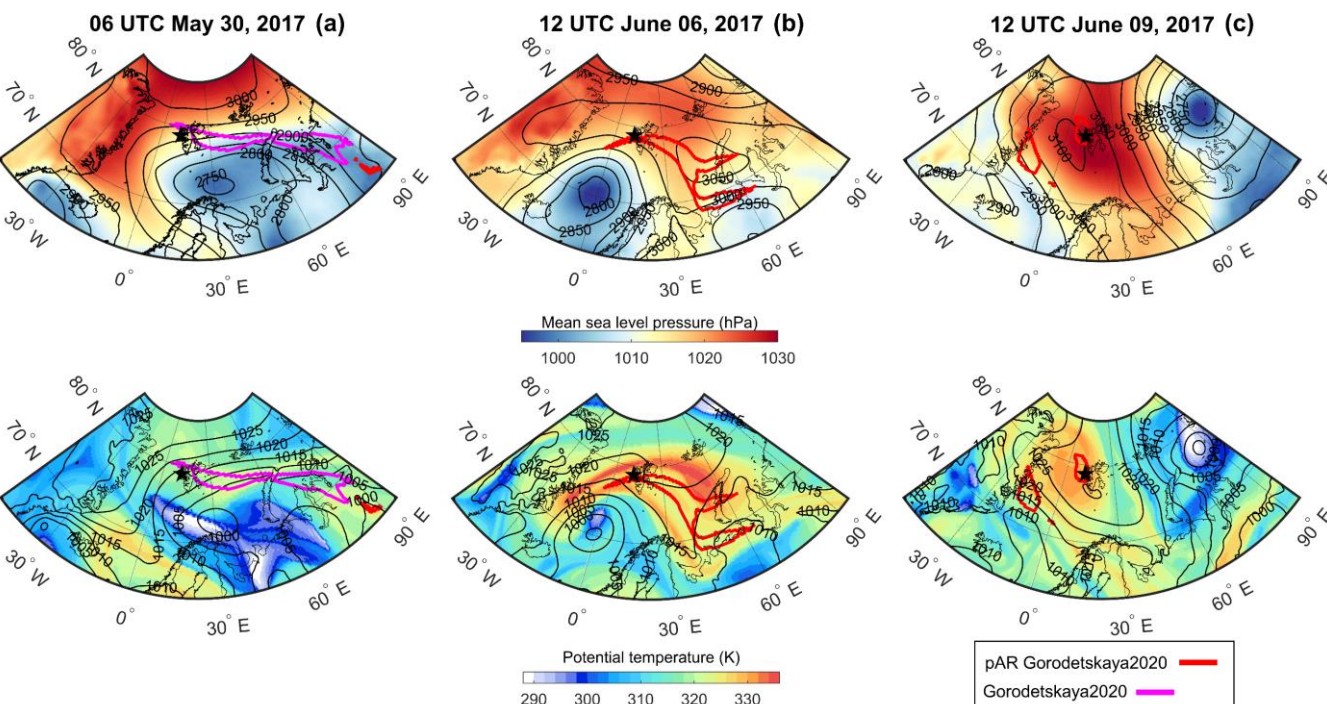

**Figure 4.** Maps of mean sea level pressure (hPa, colour shading) and geopotential height at 700hPa (m, contours) (first row) and maps of potential temperature at 2 potential vorticity units (K, colour shading) and mean sea level pressure (hPa, contours) based on ERA5 reanalysis during the peak of the 30 May event [first column, **(a)**], 6 June event [second column, **(b)**] and 9 June event [third column, **(c)**]. Magenta

line shows AR shape (based on Gorodetskaya2020) and red line shows the shape of potential ARs (IWV≥IWV$_{thres}$, based on Gorodetskaya2020). Black star shows Ny-Ålesund location. Figures S4 and S5 show the complete temporal evolution of the synoptic conditions during the events.



Three days later, a low-pressure system was located over Kara Sea, while a high-pressure system was centered over Svalbard with decreasing pressure values towards Greenland (Fig. 4c), where the AR only identified by Guan2018 was located (Fig. 2c). As previously, the pressure systems remained almost stationary, propagating slowly northeastward (Fig. S4c) leading to the curvature of the AR from northern Greenland towards northern Svalbard (Fig. S3c). In the meantime, high values of potential temperature were found from the Scandinavia Peninsula to Greenland's coast, along the shape of the AR (Fig. 4c), which were intensified in the region of the tilt of the AR towards Svalbard. These values slowly decreased with the increasing curvature of the AR towards Svalbard (Fig. S5c).

## 4.3 AR impacts at Svalbard

### 4.3.1 Variability of IWV and IVT

After analysing the spatiotemporal evolution of the events, it is also important to investigate them at a local scale. An analysis of the ARs focusing on Ny-Ålesund was performed, using all reanalysis datasets in synergy with in situ measurements (radiosonde), ground-based remote sensing (HATPRO, GNSS), satellite-based measurements (IASI L2 PPFv6) and with HIRHAM5 model. From the reanalyses and model, the nearest grid point to Ny-Ålesund is used for the comparison with the station data. The landfall time is based on the IWV peaks in Ny-Ålesund (06-12 UTC 30 May, 12 UTC 6 and 9 June).

Firstly, we assessed the temporal evolution of IWV and IVT during the events (Fig. 5). On the day before the arrival of the first event to Ny-Ålesund, the measurements, reanalyses and the model showed low IWV and IVT, which slowly increased until the beginning of the next day (Fig. 5a). During the first 6 hours, IWV continued to increase slowly. Conversely, radiosondes showed a slight decrease of IVT, which was not represented by MERRA-2 reanalysis and HIRHAM5 model. During the landfall (between 6 and 12 UTC), there was a slight increase of IWV from 11 to 15 kg m$^{-2}$, which was missed by ERA-Interim, CFSv2 and JRA-55 reanalysis, due to low temporal resolution (6 hours), along with an increase of IVT. Both IVT peaks were poorly represented by the reanalyses, with exception of ERA5. After the landfall, IVT and IWV decreased sharply, which was properly represented by all datasets.

On the day previous to the landfall of the second event, persistent low values of IWV and IVT were represented by all datasets (Fig. 5b). During the six hours before the maximum IWV occurred at Ny-Ålesund, IWV and IVT sharply increased from 6 to about 20 kg m$^{-2}$ and from 5 to more than 120 kg m$^{-1}$ s$^{-1}$, respectively (Table 2, IWV and IVT amplitude). The IWV and IVT peaks lasted around 12 hours and in the case of IWV it was misrepresented by CFSv2, JRA-55, ERA-Interim and the radiosondes, due to low temporal resolution of 6 hours. Regarding the IVT peak, a similar behaviour to the first event, with an overestimation of MERRA-2 and HIRHAM5, could be noticed (Table 2, IVT integrated during the event). IVT differences can amount up to 35 % (between ERA5 and MERRA-2) during the phase of decreasing IVT.





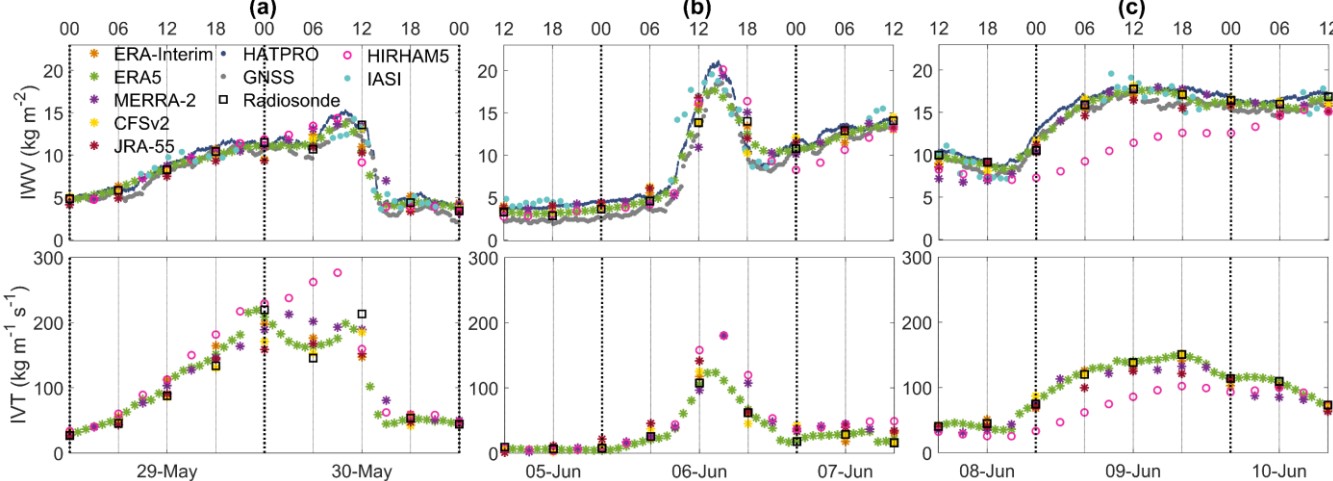

**Figure 5.** Time series of integrated water vapour (IWV, kg m$^{-2}$, first row) and integrated vapour transport (IVT intensity, kg m$^{-1}$ s$^{-1}$, second row) based on reanalyses (ERA-Interim, ERA5, MERRA-2, CFSv2 and JRA-55), radiosonde, ground-based remote sensing (HATPRO, GNSS) and satellite measurements (IASI) and HIRHAM5 model, at Ny-Ålesund, during 30 May 2017 event [first column, **(a)**], 6 June 2017 event [second column, **(b)**] and 9 June event [third column, **(c)**].

On the day prior to the third event, a slight decrease of IWV and IVT was noticed in all datasets, with exception of MERRA-2 (Fig. 5c). High values of IWV and IVT were observed during the whole day of the event, even after the landfall, although HIRHAM5 model underestimated these values by up to 55 % when compared with the radiosondes. On the following day, IVT slowly decreased, while IWV remained unchanged. Contrarily to the previous events, no prominent peak of IWV or IVT was observed, but a long duration of more than a day.

For all the events, ERA5 seems to represent more realistically the maximum and minimum values of IWV and IVT, when compared to GNSS, HATPRO and radiosondes, due to its high temporal and spatial resolution. Note that even amongst the observation datasets there are minor differences. However, previous studies showed that these differences are not significant in Ny-Ålesund with an RMSE lower than 1 kg m$^{-2}$ (Nomokonova, 2020).

During the first two events, the periods when the HIRHAM5 model overestimated the IVT might be explained by changes in the wind components, since for IWV (based on the specific humidity) the HIRHAM5 results were similar to the reanalyses and observations. An analysis of the spatial evolution of IVT based on HIRHAM5 and ERA-Interim (which was used to force the model), showed some differences in the IVT values, which were higher in HIRHAM5 model (Figs. S6 and S7). Since the edge of the band of high IVT is located around Ny-Ålesund – in the first event in the beginning (Fig. S6) and in the second event at the end (Fig. S7) – a minor difference in its location, e.g. due to slight shifts of the low and high pressure systems, induces large changes in IVT at Ny-Ålesund.




**Table 2.** Integrated water vapour (IWV, kg m$^{-2}$) and integrated vapour transport (IVT, kg m$^{-1}$ s$^{-1}$) amplitude and integrated during the event, and event duration (hours) of the AR shapes based on Gorodetskaya2020 (AR Go), of the shapes of potential ARs based on Gorodetskaya2020 (IWV≥IWV$_{thres}$) (pAR Go) and the AR shapes based on Guan2018 (only for MERRA-2 reanalysis) (AR Gu) based on reanalyses (ERA-Interim, ERA5, MERRA-2, CFSv2 and JRA-55) and HIRHAM5 model, at Ny-Ålesund, during 30 May, 6 June and 9 June 2017 events.

|  | Events | ERA-Interim | ERA5 | MERRA-2 | CFSv2 | JRA-55 | HIRHAM5 |
|---|---|---|---|---|---|---|---|
| **IWV amplitude (kg m$^{-2}$)** | 30 May | 8.1 | 9.8 | 9.9 | 9.6 | 7.7 | 11.1 |
|  | 6 June | 11.8 | 14.5 | 15.9 | 10.5 | 13.0 | 17.4 |
|  | 10 June | 8.2 | 9.2 | 11.2 | 10.0 | 7.9 | 8.2 |
| **IVT amplitude (kg m$^{-1}$ s$^{-1}$)** | 30 May | 166.3 | 179.0 | 170.3 | 152.8 | 141.1 | 242.7 |
|  | 6 June | 114.5 | 119.4 | 177.9 | 123.1 | 140.3 | 177.0 |
|  | 9 June | 99.03 | 115.3 | 101.5 | 110.1 | 85.7 | 76.9 |
| **IWV integrated during event (kg m$^{-2}$)** | 30 May | 74.5 | 70.3 | 75.0 | 70.5 | 63.0 | 70.7 |
|  | 6 June | 81.6 | 79.3 | 79.8 | 80.4 | 85.0 | 78.3 |
|  | 9 June | 125.2 | 128.2 | 124.6 | 129.4 | 121.2 | 98.3 |
| **IVT integrated during event (kg m$^{-1}$ s$^{-1}$)** | 30 May | 988.9 | 959.0 | 1003.8 | 891.9 | 874.2 | 1145.3 |
|  | 6 June | 301.7 | 280.0 | 363.1 | 312.8 | 395.9 | 452.8 |
|  | 9 June | 834.3 | 866.5 | 781.5 | 859.7 | 774.0 | 605.7 |
| **Event duration (hours)** | 30 May | pAR Go: 0 AR Go: 0 | pAR Go: 2 AR Go:0 | pAR Go: 9 AR Go: 3 AR Gu: 18 | pAR Go: 6 AR Go: 0 | pAR Go: 0 AR Go: 0 | pAR Go: 6 |
|  | 6 June | pAR Go: 0 AR Go: 0 | pAR Go: 4 AR Go: 0 | pAR Go: 6 AR Go: 0 AR Gu: 0 | pAR Go: 6 AR Go: 0 | pAR Go: 6 AR Go: 0 | pAR Go: 0 |
|  | 9 June | pAR Go: 12 AR Go: 0 | pAR Go: 14 AR Go: 0 | pAR Go: 21 AR Go: 6 AR Gu: 18 | pAR Go: 30 AR Go: 0 | pAR Go: 6 AR Go: 0 | pAR Go: 0 |

**4.3.2 Variability of vertical profiles of humidity and wind**

The vertical structure of the ARs is also an important component when studying this type of events. Figure 6 shows the vertical profiles of specific humidity and wind speed, based on reanalyses, radiosonde measurements and HIRHAM5 model, during the peaks of the events in Ny-Ålesund. The complete temporal evolution of the vertical profiles is presented in Figs. 7, S8 and





S10. For an easier comparison of the performance of each dataset, the differences between each reanalysis and model and the radiosondes are shown in Fig. S9.

During the first event on 30 May the radiosonde shows a layer of enhanced specific humidity between 1000 and 700 hPa,
which was overestimated by HIRHAM5 model (Figs. 6a, S8a and S9a). This layer was followed by a dry layer until 600 hPa only captured by the radiosonde. Wind speed values were not well represented from the surface until 650 hPa by all the reanalyses when compared with the radiosonde, as a difference in a factor of two occurs in some levels. The HIRHAM5 model largely overestimated the wind speed values along the entire column, with differences varying from 15 % at 1000 hPa, around 80 % at 850 hPa to almost 0 % at 500 hPa.

**Figure 6.** Vertical profiles of specific humidity (g kg$^{-1}$, pink/orange colours) and wind speed (m s$^{-1}$, blue/green colours) at Ny-Ålesund based on radiosonde (solid lines), reanalyses (ERA-Interim, ERA5, CFSv2, JRA-55, MERRA-2, dashed lines) and HIRHAM5 model (dotted lines), during 30 May event **(a)**, 6 June event **(b)** and 9 June event **(c)**. Figure S8 shows the complete temporal evolution of the vertical
profiles and Figure S9 shows the differences between each reanalysis and model and the radiosondes (reference).





One week later, on 6 June, with the approximation of the second event, complex vertical structure with two maxima in specific humidity of about 4 g kg$^{-1}$ at 850 hPa and 3.5 g kg$^{-1}$ at 650 hPa with a pronounced dry layer with less than 1 g kg$^{-1}$ was observed by the radiosondes. However, compared to the first event, where all datasets failed to reproduce the dry layer, the reanalyses and model show here a dry layer, but much weaker when compared to the radiosondes. It is possible that in this case, the formation of the dry layer was explained by other mechanisms which the reanalyses were able to reproduce more accurately. Furthermore, below this layer only ERA5 represented similar values of specific humidity to the radiosondes (Figs. 6b, S8b and S9b). CFSv2 and MERRA-2 are too dry and the others are too wet, and CFSv2, MERRA-2 and HIRHAM5 strongly misinterpret the vertical profile. Compared to the first event only minor differences were noticed in the wind speed, despite an overestimation of HIRHAM5 below the 850 hPa. Six hours later, the dry layer was still present with even lower values of specific humidity and its base moved upwards (Fig. S8b). A study performed by Neggers et al. (2019) analysed data from radiosondes launched from the Polarstern research vessel during the period of 5 to 7 June 2017. In this study, similar dry layers were identified in western Svalbard during June 6 at 04 and 10 UTC, around 2.5 km and 2 km height, respectively.

Three days later, during the third event, on 9 June, the radiosondes captured a layer with high values of specific humidity up to 5 g kg$^{-1}$ below 800 hPa, which was represented by all reanalysis datasets (Figs. 6c, S8c and S9c). HIRHAM5 model largely underestimated the specific humidity until 600 hPa and showed an unrealistic decrease of humidity with height. The wind speed profiles were properly represented by all datasets with the calmest situation of all events in the lower tropopause.

The vertical profiles are in agreement with Fig. 5, since the reanalyses/model overestimation (underestimation) of specific humidity in some or all vertical levels lead to higher (lower) values of IWV. Furthermore, the overestimation of HIRHAM5 wind speed during the first two events, mainly near the surface, and differences in the amounts of specific humidity might explain the major differences in HIRHAM5 IVT noticed in Figs. S6 and S7. Also, the underestimation of HIRHAM5 specific humidity in the last event, explains the major differences in IWV and IVT observed in Fig. 5c.

The temporal evolution of specific humidity vertical profiles during the three events based on radiosondes, reanalyses and HIRHAM5 model is illustrated in Fig. 7. On the day prior to the first event the radiosondes show low values of specific humidity. Associated with the approaching event, specific humidity showed a sharp increase, with a moist layer extending from the surface until 675 hPa with a peak around 800 hPa. These observations also captured a dry layer present above the moisture peak (from 700 to 600 hPa) at 06 UTC. Maximum moisture values were observed at 12 UTC followed by a sharp decrease. Overall, the height of the maximum increase in specific humidity (around 675 hPa) was well captured by all the reanalyses and model, although MERRA-2 and CFSv2 (Fig. S10a) showed a more extended layer of moist air. Furthermore, JRA-55 and ERA-Interim showed the lower amounts of specific humidity in the peak of the moisture layer (800 hPa) (Fig. S10a). The dry layer was not properly represented by any of the reanalyses and model which might be due to its narrow vertical extent of about 85 hPa. It is also interesting to note that the observed reduced moisture within the whole column after the event is not fully realistic in the reanalyses; only ERA5 and HIRHAM5 showed the reduction also in the low layers near the surface.

One week later, a stronger moisture intrusion associated to the second event reached Ny-Ålesund. Before its approach, low specific humidity values were found, followed by an intense and rapid increase of moisture. Before the peak of the event there

**Figure 7.** Temporal evolution of vertical profiles of specific humidity (g kg⁻¹) based on radiosondes, reanalyses (ERA-Interim, ERA5, MERRA-2), and HIRHAM5 model, during 30 May 2017 event [first row, **(a)**], 6 June 2017 event [second row, **(b)**] and 9 June event [third row, **(c)**], at Ny-Ålesund. Time steps on the x-axis mark the end of observations/reanalysis/model. Figure S10 shows all datasets.



was a moist layer from the surface until 800 hPa, below a dry layer, which extended until 700 hPa. At the peak this moist layer extended upward until 750 hPa, followed by a sharp decrease of the moisture amount. By the end of the day, high amounts of specific humidity were still captured below 850 hPa in Ny-Ålesund. The reanalyses and model represented well the timing and height of the elevated moisture intrusion associated with the event. Overall, the amount of specific humidity was well represented by the reanalyses and model, despite the underestimation of ERA-Interim, CFSv2 and JRA-55 (Fig. S10b).

However, the dry layer was not captured well by the reanalyses or model, with exception of the highest resolution reanalysis ERA5 which shows the moisture inversion, despite its intensity was strongly underestimated.

Two days later, high amounts of specific humidity were captured by the radiosondes below 800 hPa. On the following day, with the arrival of the third event, the moisture amounts increased accompanied by the expansion of the height of the maximum specific humidity until 650 hPa. After the event, high amounts of humidity were still noticed, and the height of maximum

specific humidity remained unchanged until the following day. The reanalyses properly represented the height of maximum specific humidity, despite underestimating the amounts of specific humidity. The more pronounced differences were noticed in HIRHAM5 model, which misrepresented the height of maximum humidity, and the amount of specific humidity was underestimated.

### 4.4 Precipitation patterns during the ARs

ARs are usually associated to intense precipitation events, which in the Arctic might occur in the form of rain or snow. Prior studies have associated extreme precipitation events to ARs in Svalbard (Kelder et al., 2020), with nearly half of the 15 largest precipitation events in Ny-Ålesund from 1979 to 2014 due to ARs (Serreze et al., 2015). Evidence for the influence of ARs in the sea-ice loss to the Arctic region has also been shown (Wang et al., 2020). However, an assessment of the sea-ice retreat or expansion mechanisms is beyond the scope of this study.

In this section, we performed a spatial analysis of the precipitation patterns related to the identified ARs, and associated changes in the sea-ice edge, using reanalysis data, remote sensing measurements and the HIRHAM5 model (Fig. 8). The analysis was completed by the discrimination of the precipitation phase, in terms of snowfall and rainfall (Figs. 9, S11, S12 and S13). This analysis was based on the accumulated amounts of precipitation during 48 hours periods (24 hours before and after the events reached Ny-Ålesund). A similar procedure was applied to the outline of the ARs that were previously identified by the tracking

algorithms. Thus, the total AR shapes shown in Figs. 8, 9, S11, S12 and S13 correspond to the total area occupied by each pAR/AR shape during the 48 hours periods (similar to precipitation), as these shapes moved and evolved during each event.

During the first event all reanalyses show an enhanced band of precipitation within the pAR/AR shape from Western Siberia to Barents Sea (Figs. 8a and S11a). However, ERA-Interim and ERA5 show localized high values of precipitation (>25 mm accumulated during 48 hours) in Russia's mainland and in the northern Novaya Zemlya and adjacent region of Kara Sea, while

MERRA-2, CFSv2 and JRA-55 reanalyses show a continuous band of high amounts of precipitation (maximum total precipitation values >40 mm during 48 hours) from Western Siberia extending through Kara Sea until Novaya Zemlya (Figs. 8a and S11a). Simultaneously, HIRHAM5 model has a similar pattern to the reanalyses but high values of precipitation are

**Figure 8.** Maps of the total accumulated precipitation (mm, colour shading) for the 30 May event [first column, **(a)**], 6 June event [second column, **(b)**] and 9 June event [third column, **(c)**] during a 48 hours period (24 hours before and after AR reaches Ny-Ålesund, shown by black star) based on reanalyses (ERA-Interim, ERA5, MERRA-2) and HIRHAM5 model. Grey lines show the sea-ice fraction using a 15 % threshold (thin line represents 24 hours before the event and thick line 24 hours after the event). Magenta and red lines show the AR and pAR shapes, respectively, based on Gorodetskaya2020. Black line shows the AR shape based on Guan2018 (available only for MERRA-2). The AR shape lines here encompass the total area of the ARs/pARs during the 48 hours period. Figures S11, S12, S13 show the discrimination of the precipitation phase (snowfall and rainfall) for all datasets.



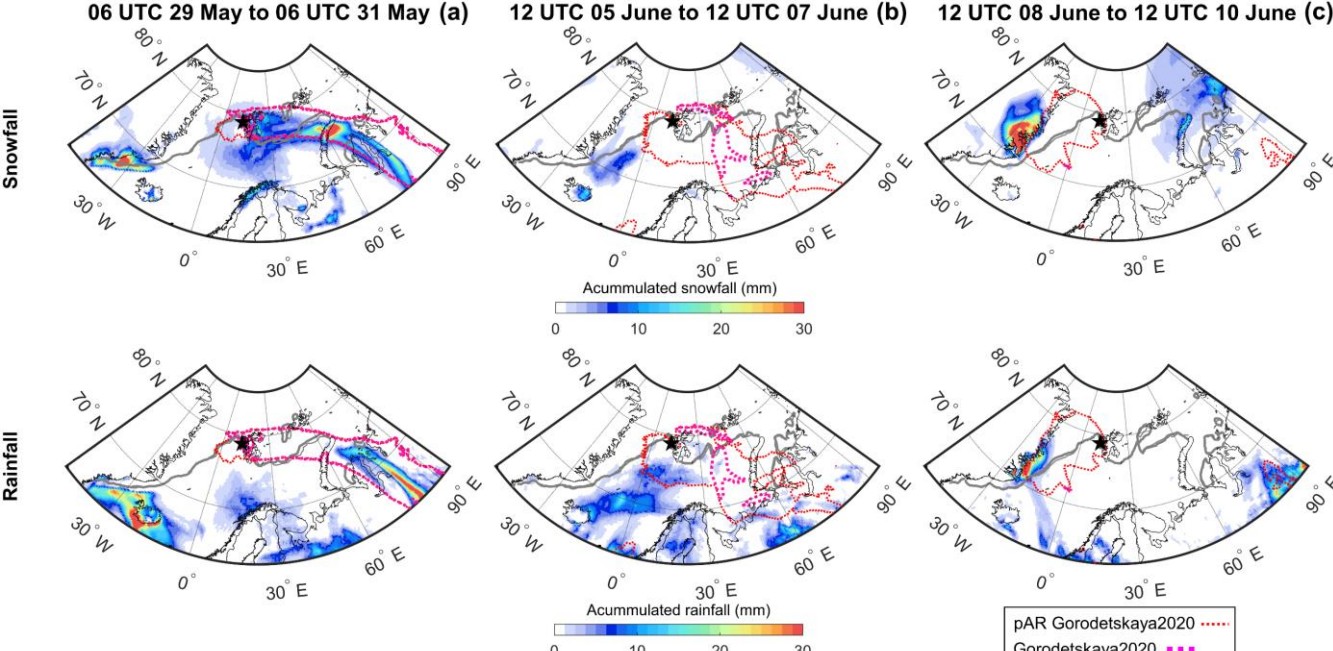

**Figure 9.** Maps of the accumulated snowfall (mm, colour shading, first row) and rainfall (mm, colour shading, second row) for the 30 May
event [first column, **(a)**], 6 June event [second column, **(b)**] and 9 June event [third column, **(c)**] during a 48 hours period (24 hours before
and after the AR reaches Ny-Ålesund, shown by the black star) based on ERA5 reanalysis. Grey lines show the sea-ice fraction using a 15
% threshold (thin line represents 24 hours before the event and thick line 24 hours after the event). Magenta and red lines show the AR and
pAR shapes, respectively, based on Gorodetskaya2020. The AR shape lines here encompass the total area of the ARs/pARs during the 48
hours period. Figures S11, S12, S13 show the discrimination of the precipitation phase (snowfall and rainfall) for all datasets.

restricted to Kara Sea and northern Novaya Zemlya (maximum precipitation of 90 mm during 48 hours). This island,
characterized by its high orography mainly in the northern latitudes (maximum ~1500 m), caused orographic enhancement of
precipitation. Thus, from this island towards Svalbard precipitation amounts were reduced, despite, depending on the reanalysis
dataset, some smaller amounts of precipitation (<10 mm accumulated during 48 hours) were noticed in HIRHAM5 model, and
might be mainly related to the Foehn effect. Compared to the precipitation climatology of Svalbard region (from 1979 to 2018)
that varies from 31 mm in Svalbard Airport station to 127 mm in Barentsburg station, and 89 mm in Ny-Ålesund station,
accumulated during Spring (March to May) (Førland et al., 2020), the amount of precipitation reaching this region during the
event is small. If we look in more detail to a monthly climatology of May (from 1951 to 1980), where in Barentsburg station
precipitation was 25 mm during an average of 14 days (Aleksandrov et al., 2005), the amount of precipitation associated to the
AR (accumulated during 2 days) was around 1 mm, which is reduced compared to the climatological amounts. The same
climatology, but for Zhelaniya Cape station (northern region of Novaya Zemlya), shows that the average precipitation during
May was 23 mm during 17 days (Aleksandrov et al., 2005), which compared with the precipitation amounts verified in this
region during the AR (in some datasets >10 mm during 2 days) shows that this event was significant depending on the dataset





(for MERRA-2 11.7 mm during 2 days, corresponding to ~50 % of the monthly climatological precipitation; for ERA5 0.3 mm during 2 days). For Dikson Island station (located in northern Russia), the climatology shows an average precipitation of 26 mm during 18 days in May (Aleksandrov et al., 2005), while during the AR, precipitation amounts reached 11.5 mm (MERRA-2 reanalysis) during 2 days, representing a significant amount of precipitation in this region (44 % of the climatological monthly precipitation). However, it is important to note that in ERA5 reanalysis, precipitation in this region

was around 0.6 mm.

    The majority of the precipitation was confined to the AR shape, but in southern Svalbard precipitation also occurred outside the AR associated with the extra-tropical cyclone. During this AR event, precipitation occurred as mixed-phase, showing both snow and rain within the AR, and major differences were noticed across the reanalyses and model (Figs. 9a, S11b and S11c). ERA-Interim and JRA-55 were the only datasets with rainfall in Svalbard region, while CFSv2, MERRA-2 and ERA5

reanalyses and HIRHAM5 model only show rainfall in Western Siberia and in the adjacent coastal region (Fig. S11b). CFSv2 shows the highest amount of rainfall (>25 mm accumulated during 48 hours) along Western Siberia and a small portion of Kara Sea during the 48 hours period, whereas ERA-Interim, MERRA-2, ERA5 and JRA-55 only have this amount of rainfall in the inner Western Siberia region, and HIRHAM5 model does not even show such values of rainfall in this region (Fig. S11b). Simultaneously snowfall reached regions further north, extending from Western Siberia towards Svalbard (Fig. S11c).

The higher amounts of snowfall were noted in the Kara Sea and Novaya Zemlya (>40 mm during 48 hours, with exception of ERA-Interim reanalysis), with their accentuated decrease northwestward of this island. The highest amounts of snowfall and rainfall were found in the southern part of the AR and south of the sea-ice edge, in all datasets. However, smaller amounts of snowfall occurred over the sea-ice, while rainfall was confined to regions south of the sea-ice edge, over the open-sea, with exception of ERA-Interim and JRA-55.

A full analysis of the total and mean precipitation amounts and discrimination of precipitation phase within the pAR shape by Gorodetskaya2020, and AR shapes by Gorodetskaya2020 and Guan2018 is shown in Table 3. Overall, the area of the AR and pAR shapes was similar across all the reanalyses. The exception was the AR shape by Guan2018, based on MERRA-2 reanalysis, which was more than two and three times larger than the pAR and AR shapes by Gorodetskaya2020, respectively, due to the different criteria used by the algorithms. These was associated to higher total amounts of total precipitation, mostly

in the form of rainfall. Furthermore, one can notice that the AR shapes by Gorodetskaya2020 have higher mean values of total precipitation, rainfall and snowfall than the pAR shapes, which is explained by the AR shapes being more restrictive, containing only higher amounts of precipitation. In particular, CFSv2 and JRA-55 show the higher values of total and mean total precipitation, mainly due to higher values of rainfall when compared with the remaining reanalyses.

    The sea-ice edge showed a retreat, mainly in the northern Barents Sea region north of Novaya Zemlya (Figs. 8a and S11),

which might be explained by different mechanisms, such as high wind speed with a northwestward direction in the region of Novaya Zemlya, as mentioned previously (Figs. 2a, S1a and S6). This intense wind blowing over the limit of the sea-ice edge, which is defined by areas with at least 15 % ice cover meaning it is already fragile, might have pushed the sea-ice further north, causing its retreat.



**Table 3.** Total and mean total precipitation, snowfall and rainfall amounts within the pAR/AR shapes by Gorodetskaya2020 and Guan2018 (mm during 48 hours) based on reanalyses (ERA-Interim, ERA5, MERRA-2, CFSv2 and JRA-55) during the 30 May 2017 event.

| | Shapes | Shapes area ($\times10^6$ km$^{-2}$) | Total precipitation (mm/48 hours) | | Snowfall (mm/48 hours) | | Rainfall (mm/48 hours) | |
|---|---|---|---|---|---|---|---|---|
| | | | Mean | Total ($\times10^7$) | Mean | Total ($\times10^7$) | Mean | Total ($\times10^7$) |
| **Era-Interim** | pAR Gorodetskaya2020 | 2.1 | 10.4 | 2.2 | 2.5 | 0.5 | 7.9 | 1.7 |
| | AR Gorodetskaya2020 | 1.6 | 12.0 | 1.9 | 3.3 | 0.5 | 8.7 | 1.4 |
| **ERA5** | pAR Gorodetskaya2020 | 2.6 | 9.6 | 2.5 | 3.5 | 0.9 | 6.1 | 1.6 |
| | AR Gorodetskaya2020 | 2.2 | 10.5 | 2.3 | 4.1 | 0.9 | 6.4 | 1.4 |
| **MERRA-2** | pAR Gorodetskaya2020 | 2.9 | 11.8 | 3.4 | 5.8 | 1.7 | 6.0 | 1.7 |
| | AR Gorodetskaya2020 | 2.3 | 12.7 | 2.9 | 7.0 | 1.6 | 5.7 | 1.3 |
| | AR Guan 2018 | 7.4 | 6.5 | 4.8 | 2.5 | 1.8 | 4.0 | 2.9 |
| **CFSv2** | pAR Gorodetskaya2020 | 2.5 | 14.2 | 3.5 | 5.2 | 1.3 | 9.0 | 2.2 |
| | AR Gorodetskaya2020 | 1.8 | 15.2 | 2.8 | 6.0 | 1.1 | 9.2 | 1.7 |
| **JRA-55** | pAR Gorodetskaya2020 | 2.0 | 16.1 | 3.2 | 5.1 | 1.0 | 11.0 | 2.2 |
| | AR Gorodetskaya2020 | 1.5 | 15.8 | 2.3 | 6.7 | 1.0 | 9.1 | 1.3 |

Despite the similarities to the first event described in the previous sections, the second event, only one week later, on June 6, was completely different in terms of precipitation patterns, with low amounts of precipitation within the AR shape (< 15 mm accumulated over the 48 hours period) (Figs. 8b and S12a). The majority of precipitation occurred southwest of the AR shape, directed towards Iceland, although in CFSv2 reanalysis precipitation occurred partially within the pAR shape and in MERRA-2 reanalysis the AR shape by Guan2018 extended more towards Iceland, partially including precipitation, which was also noticed in the pAR shape by Gorodetskaya2020 in HIRHAM5 model, including precipitation from this region (Figs. 8b and S12a). All reanalyses and HIRHAM5 model show similar total precipitation patterns, although ERA-Interim has the lowest amounts of precipitation (maximum of 15 mm during 48 hours south of the pAR/AR shapes). Most of the total precipitation occurred in the form of rain (Figs. 9b and S12b), with exception of some areas where mixed-phase precipitation or only snowfall occurred (Figs. 9c and S12c), mainly near the Greenland's coast and in the northeast of Iceland. Precipitation mainly occurred over ice-free ocean with the exception of the area in south and east of Svalbard, where low values of rainfall were noted (< 5 mm accumulated during 48 hours), simultaneously with reduced amounts of snowfall (< 10 mm accumulated during 48 hours) near Greenland's coastline. No differences were noted in the sea-ice edge, possibly due to the reduced amounts of precipitation over the sea-ice (rain or snow) and low values of IVT, and consequently wind speed, over the sea-ice (Fig. S7).





During the third event, three days later, on June 9, no precipitation was noticed in Svalbard, which was located at the edge of the pAR/AR (Fig. 8c). At the same time, high amounts of precipitation occurred in the east coast of Greenland, in the mountainous region of Scoresby Land (> 20 mm accumulated during 48 hours period) confined within the AR shape defined

by Guan2018 and in the edge of the pAR shape defined by Gorodetskaya2020 algorithm. In this region, total precipitation amounts were similar in all reanalyses and HIRHAM5 model (Fig S13a), however the discrimination of the precipitation phase shows major differences (Figs. S9c, S13b and S13c). With the exception of MERRA-2, all datasets show high amounts of rainfall in the coastal region of Greenland, over the sea-ice (maximum of 64 mm and 110 mm during 48 hours in JRA55 reanalysis and HIRHAM5 model, respectively), together with high amounts of snow in the adjoining continental area

(maximum of 80 mm and 200 mm during 48 hours in CFSv2 reanalysis and HIRHAM5 model, respectively). MERRA-2 presents low values of rainfall in coastal Greenland (maximum of 11 mm during 48 hours) and high amounts of snow in the continental and coastal regions (maximum of 75 mm during 48 hours). As observed in the last event, there were no major changes in the sea-ice extent.

A previous study by Boisvert et al. (2018) pointed to major differences in precipitation amount and phase over the Arctic

Ocean between eight reanalyses datasets, in which ERA-Interim, MERRA-2, JRA-55 (analysed in our study) are included. The largest annual differences were found in east Greenland, Kara and Barents Sea, which might be explained by the influence of storm track and how the reanalyses assimilate those events. The monthly analysis of the cumulative precipitation during May and June over this region shows no major discrepancies between the three reanalyses used in our study. The discrimination between snowfall and rainfall showed big differences amongst the reanalyses. As observed in our study, MERRA-2 showed

higher amounts of snowfall over the Barents and Kara Seas and coastal Greenland in comparison with ERA-Interim and JRA-55. The variability of rainfall between reanalyses is bigger along the east coast of Greenland and, as in our study, MERRA-2 has the lowest amounts of rainfall compared to the other reanalysis.

Finally, we performed an analysis of the air mass trajectories during the AR events using HYSPLIT model (Fig. S14). The start date to calculate an ensemble of the back trajectories for the 5 previous days was defined based on the IWV peaks in Ny-

Ålesund (06 UTC 30 May, 12 UTC 6 and 9 June). The trajectories were initiated at 800 hPa height at the location of Ny-Ålesund. During the first AR the trajectories showed low variance until 24 hours previous to the initial date, with a mean trajectory path over the Barents and Kara Seas before reaching Ny-Ålesund site. Over the continent the trajectories showed a higher variability with the majority being confined to Western Siberia (Fig. S14 – left panel). The second event showed that the majority of trajectories passed over the Kara and Barents Seas before reaching Svalbard. After reaching the continent over

the Western Siberia, some trajectories passed over the Baltic Sea, but the majority were limited to the region west of the Ural Mountains (Fig. S14 – middle panel). Few trajectories showed Greenland and northern Canada as a possible air mass path. The last event had a distinct behaviour, with the air mass trajectories passing over the Norwegian and Greenland Seas before reaching Ny-Ålesund (Fig. S14 – right panel). The trajectories extend over the Norwegian Sea towards the North Sea. Here we show only air mass trajectories and further analysis of the moisture sources and links to precipitation patterns is needed in

order to investigate possible moisture uptake along the trajectory of the ARs over time (beyond the scope of this study).



## 5 Summary and conclusions

This study comprises the analysis of three anomalous water vapour transport events in the Arctic identified during the ACLOUD/PASCAL campaigns, which took place from May 22 to June 28, 2017, at and near Svalbard. Five reanalysis products (ERA5, ERA-Interim, MERRA-2, CFSv2, JRA-55) were used to analyse the events and compared with the
measurements at the AWIPEV research station (in Ny-Ålesund; HATPRO, GNSS and radiosondes), satellite-borne measurements (IASI) and a regional climate model intensively used for Arctic climate studies (HIRHAM5). The events took place on 30 May, 6 and 9 June 2017 and were identified as atmospheric rivers by either one or both AR algorithms by Gorodetskaya2020 and Guan2018. These AR events explained three out of four anomalous values of IWV and IVT observed at Ny-Ålesund during the duration of the ACLOUD/PASCAL campaigns.

The first AR event reaching Svalbard on 30 May was associated to a band with high values of IWV extending from Western Siberia to Svalbard. The impacts of this event included a band of enhanced mixed-phase precipitation, showing both snow and rain confined to the AR shape. Although snowfall occurred over the sea-ice, the higher amounts occurred south of the sea-ice edge, while rainfall was confined to the open-sea. Concurrently, a retreat of the sea-ice extent was noticed mainly in the Barents Sea, which might be explained by high wind speed in this region. One week later, on June 6, the second AR event affected
Svalbard and was composed of two bands of enhanced moisture extending from Western Siberia, converging into one wider filament near Novaya Zemlya, with an outstanding zonal component. This event caused low amounts of precipitation, mainly southwest of the AR shape, in the form of rain over the ice-free portion of the ocean, associated to no major differences in the sea-ice edge. This AR event with a predominant zonal component was detected as potential AR by Gorodetskaya2020 algorithm and was not detected as an AR by the global Guan2018 algorithm, where the meridional poleward moisture is
emphasized. Following these results, current work aims at adapting the Gorodetskaya2020 algorithm in order to include ARs with a strong zonal component and reduced meridional component. Three days later, on 9 June, the third AR event extended from northeastern Atlantic towards Greenland, turning southeastward reaching Svalbard, with a strong meridional component. This event caused no precipitation in Svalbard, although high amounts of precipitation occurred in the coast of Greenland, with snow and rain confined to the continental and coastal regions. No major changes in the sea-ice extent were found during
this event.

The five reanalysis products and HIRHAM5 model represented properly the spatial IWV patterns when compared with satellite measurements (IASI L2 PPFv6). However, the horizontal and temporal resolution of the reanalysis fields, and the physical parametrizations of the model as well as data assimilation (Rinke et al., 2019) can have a determinant role on the identification and shape of the AR. Furthermore, total precipitation amounts were distinct amongst the five reanalyses and HIRHAM5 model,
together with major differences in the discrimination of the precipitation phase. A study by Boisvert et al. (2018), which included the analysis of precipitation based on eight different reanalysis products from 2000 to 2016, also pointed to discrepancies in the precipitation phase.



Following the spatial analysis of the ARs, we investigated their impacts at Ny-Ålesund (Svalbard), particularly in the temporal evolution of IWV and IVT and the vertical structure of the ARs, based on the profiles of specific humidity and wind speed. Overall, the temporal evolution of IWV and IVT was properly represented by the reanalyses and HIRHAM5 model. Differences were found in the IWV during the first and second events, where ERA-Interim, CFSv2 and JRA-55 reanalyses missed the peaks, due to low temporal resolution, concurrently with an overestimation of IVT by MERRA-2 and HIRHAM5. During the third event, both IWV and IVT were underestimated by HIRHAM5. IWV and IVT values differed significantly depending on the event. The mean maximum and minimum values of IWV and IVT (based on the five reanalysis) during the 30 May event ranged from 3 to 13 kg m$^{-2}$ and 30 and 196 kg m$^{-1}$ s$^{-1}$, during the 6 June event varied from 3 to 17 kg m$^{-2}$ and 2 and 137 kg m$^{-1}$ s$^{-1}$ and during the 9 June event fluctuated from 8 to 17 kg m$^{-2}$ and 37 and 140 kg m$^{-1}$ s$^{-1}$. Focusing on the vertical profiles of specific humidity, the radiosondes identified layers of enhanced moisture, which were well represented by the reanalyses, simultaneously with dry layers during the first two events, which were not captured by all reanalysis datasets. HIRHAM5 overestimated humidity during the first two events, while during the third event the specific humidity was largely underestimated. Regarding the wind speed, the first and last events showed an increase of values from the lower to upper layers, while during the second event there were no major changes in the wind speed with height, but a low-level wind jet formed. In the first event wind speed was misrepresented by all reanalyses and HIRHAM5, while in the second event there was a decrease in these differences and in the third event all reanalyses and HIRHAM5 represented well the wind speed. For all the events, ERA5 seems to represent more appropriately the maximum and minimum values of IWV, IVT and vertical profiles of specific humidity and wind, when compared to the reference datasets (GNSS, HATPRO and radiosondes), due to its high temporal and spatial resolution.

Concluding, during a short period of time (less than two weeks) three intense and short duration AR events affecting Svalbard were identified. Despite being consecutive they had different moisture amounts and transport, vertical structure, precipitation amounts and phase, and moisture sources. Although the results show a reasonable comparison between the reanalysis datasets, a regional climate model and in situ and remote sensing measurements, this study shows the importance of using datasets with the appropriate spatial and temporal resolution when assessing extreme short duration events, such as ARs. The temporal and/or spatial resolution of the reanalysis datasets and measurements directly influences both IWV and IVT and consequently the identification of ARs. Thus, one should use reanalyses and model simulations with high spatial and temporal resolution, such as ERA5, along with measurements obtained during short time intervals.

In this study we focused in understanding the mechanisms of ARs in the Arctic and their relation with changes in moisture amounts and precipitation in this region. As a future work, we plan to extend this analysis to longer time periods from the historical to future periods, using reanalyses and global climate models, in order to understand their importance and magnitude in terms of moisture transport and associated precipitation amounts, due to climate change.



## 6 Data availability

The in situ and ground-based remote sensing measurements used in this paper, are available in PANGAEA: radiosondes (https://doi.pangaea.de/10.1594/PANGAEA.879820; https://doi.pangaea.de/10.1594/PANGAEA.879822) and HATPRO (https://doi.pangaea.de/10.1594/PANGAEA.902142).

The satellite data used in this study is available in EUMETSAT – IASI (https://archive.eumetsat.int/usc/). GNSS data was provided by the GeoforschungsZentrum Potsdam (GFZ).

The reanalysis datasets used in this study were provided by ECMWF: ERA-Interim (https://www.ecmwf.int/en/forecasts/datasets/reanalysis-datasets/era-interim) and ERA5 (https://www.ecmwf.int/en/forecasts/datasets/reanalysis-datasets/era5); NCEP: CFSv2 (https://rda.ucar.edu/datasets/ds094.0/); JMA: JRA-55 (https://rda.ucar.edu/datasets/ds628.0/); NASA: MERRA-2 (https://goldsmr4.gesdisc.eosdis.nasa.gov/data/MERRA2/).

HIRHAM5 model data are available at the tape archive of the German Climate Computing Center (DKRZ; 675 https://dkrz.de/up/systems/hpss/hpss); one needs to register at DKRZ to get a user account. We will also make the data available via Swift (https://www.dkrz.de/up/systems/swift) on request.

## 7 Code availability

Guan2018 AR tracking algorithm is provided by Bin Guan via https://ucla.box.com/ARcatalog. Gorodetskaya2020 algorithm is available upon request (contact: irina.gorodetskaya@ua.pt). Both algorithms are part of the ARTMIP 680 (https://www.cgd.ucar.edu/projects/artmip).

*Author contributions.* CV and IG led the coordination and design of the study and interpretation of the results. AnR, MM, SC provided the datasets used in the study. CV processed the data, plotted the figures and drafted the manuscript with IG input. All co-authors contributed to the editing and revising of the manuscript. IG developed the AR algorithm codes and both CV and IG adapted it to the Arctic.

*Competing interests.* The authors declare that they have no conflict of interest.

*Acknowledgments.* This work is supported by Fundação para a Ciência e Tecnologia (FCT) (Portugal), by the PhD Grant reference SFRH/BD/129154/2017. Thanks are due to FCT/MCTES for the financial support to CESAM (UIDP/50017/2020+UIDB/50017/2020), through national funds. The authors gratefully acknowledge the funding by the Deutsche Forschungsgemeinshaft (DFG, German Research Foundation) – Projektnummer 268020496 – TRR 172, within the 690 Transregional Collaborative Research Center "ArctiC Amplification: Climate Relevant Atmospheric and SurfaCe Processes, and Feedback Mechanisms (AC)3". We also wish to thank Alfred Wegener Institute and University of Cologne. The authors further acknowledge the NOAA Air Resources Laboratory (ARL) for the provision of the HYSPLIT transport and dispersion





model and/or READY website (http://www.ready.noaa.gov) used in this publication. We thank B. Guan for providing his algorithm data (https://ucla.box.com/ARcatalog).

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
