# Peer review of "Atmospheric rivers and associated precipitation patterns during the ACLOUD/PASCAL campaigns near Svalbard (May-June 2017): case studies using observations, reanalyses, and a regional climate model"

_Atmospheric Chemistry and Physics, 2021_

## Author Comment (AC1)

**Comment on acp-2021-609**

**Anonymous Referee #2**

**General comments:**

This paper explores the differences of atmospheric rivers' characteristics and meteorology in reanalyses and the regional climate model HIRHAM5 in Svalbard. Using field campaign identification of water vapor transport events in May and June 2017, independent atmospheric river events were detected at a research station over an 11-day period. This paper analyzes how using integrated vapor transport versus integrated water vapor affects atmospheric river detection. Additionally, the differences between reanalyses, the regional climate model, and observations demonstrate the importance of using a high spatial and temporal resolution model for AR identification. This paper was a thorough study in how applications of AR detection schemes vary in the Arctic, where lower atmospheric moisture content must be considered. Additionally, the synoptic and precipitation analyses were very interesting, particularly how the results varied amongst the reanalyses. There were some points that needed clarification, but overall, the manuscript was scientifically interesting, novel, and important work. For these reasons, I am recommending this manuscript for acceptance to ACP with minor revisions.

The authors thank the referee for taking the time to carefully review the manuscript. We believe the manuscript will benefit from these revisions. Below we addressed all the questions raised by the referee. Comments from the referee are in **black** and the responses from the authors are in **blue**.

**Specific comments:**

Section 3.2: Some added clarification on which datasets the algorithms were applied to would be helpful. The final two sentences (L208-210) specifically state that the Guan2018 algorithm was only applied to MERRA-2, but I was confused until getting into the results about which datasets Gorodetskaya2020 was applied to. It is not stated until L230-231.

The authors agree that due to the several reanalysis products and algorithms used in this manuscript it is important to include a clarification about their application. Gorodetskaya et al. (2020) was applied to ERA-Interim, ERA5, CFSv2, JRA-55, MERRA-2 and HIRHAM5. This additional information was included in a new sentence in the updated version of the manuscript:

"ERA-Interim, ERA5, CFSv2, MERRA-2 and JRA-55 reanalysis were used to identify pARs/ARs, while HIRHAM5 model was only used to identify pARs, due to its limitation on spatial coverage (approximately, northern 65° N), both based on Gorodetskaya2020."

Also, the sentence about which datasets were applied to Guan et al. (2018) was improved:

"Only MERRA-2 reanalysis, covering a period from 1980 to 2019, were used to calculate IVT and to detect the ARs based on Guan2018."

**L 228-230: Why were these specific datasets chosen to display in Figure 2, when you also applied Gorodetskaya2020 to JRA55 and CFSv2?**

With the purpose to keep Figure 2 as simple as possible, for the maps to have a reasonable size, and to compare the three events, we decided to just show the maps for some datasets and only for the time of maximum IWV. Figures S1, S2 and S3 show all datasets, the timing previous, during and after the maximum IWV, and a wider map with more longitudes.

For Figure 2, we decided to only include results from ERA-Interim, ERA5 and MERRA-2 reanalyses, HIRHAM5 model and IASI observations. The choice of these datasets was done in a way to compare the reanalyses, model (HIRHAM5) and observations (IASI). Concerning the reanalyses, we chose ERA-Interim because it is one of the most widely used datasets, while ERA5 has the higher spatial and temporal resolution. Regarding MERRA-2, it was important to include this dataset in this figure in order to compare results from Gorodetskaya et al. (2020) algorithm and Guan et al. (2018) algorithm (only applied on MERRA-2 reanalysis).

L 242-244: While it is later elaborated on in L272-277, it might be helpful to mention, after stating that the geometrical criteria was applied, that the current geometrical applications of the algorithm prevent the June 6 case from being identified as an AR (and is instead a pAR).

The authors thank the suggestion from the referee. This information was included in the new version of the manuscript, referring to a more detailed explanation later in the document:

"After applying Gorodetskaya2020 tracking algorithm, two of the four events were detected as pARs: 30 May and 6 June (Figs.2a and 2b, red lines; Figs. 3a and 3b, coloured circles). With the inclusion of the geometrical criteria only the first event was identified as an AR (Fig. 2a, magenta line; Fig. 3a, coloured dots), since the current geometrical criteria prevent the 6 June event to be identified as an AR (explained later)."

**Figure 3: Yellow color is difficult to see - could this be changed to a darker yellow?**

Figure 3 was updated in order to be more visible. The yellow colour, previously used to represent the results from CFSv2, was changed to blue. Furthermore, the display of the figure was changed since referee #3 suggested the figure was not attractive. The

updated figure is shown below and it was included in the revised version of the manuscript.

The authors thank the comment from the referee. We noticed that the sentence was not clear. In here, we are not referring to time, but to location. When we refer to *"in the first event in the beginning (Fig. S6) and in the second event at the end (Fig. S7)"*, our purpose was to mention the position of the band of high IVT in relation to Ny-Ålesund. In the case of the 30 May event (Figure S6), the beginning of the edge of the band with high IVT is located near Ny-Ålesund, while in the 6 June event (Figure S7) it is the end of the edge of the band with high IVT that is located near Ny-Ålesund.

The sentence was updated in the new version of the manuscript, in order to be clearer:

"Since in the first event the beginning of the edge of the band of high IVT is located around Ny-Ålesund (Fig. S6) and in the second event it is the end of this band located near Ny-Ålesund (Fig. S7), a minor difference in its location, e.g. due to slight shifts of the low and high pressure systems, induces large changes in IVT at Ny-Ålesund."

**Figure 7: These figures could probably be made wider by cutting whitespace on the edges to make the timestamps more clear.**

The plots from Figure 7 were changed in order to get wider and more visible for the readers. The space between plots and their height was reduced. The new version of the figure is shown below:

---

## Author Comment (AC2)

**Comment on acp-2021-609**

**Anonymous Referee #3**

**General comments:**

The paper by Viceto et al. 2021 analyses three atmospheric river events which took place during the ACLOUD/PASCAL campaigns near Svalbard. They compare model simulations, with a wide variety of reanalysis datasets and observations. In addition, they use two different detection methods for Atmospheric rivers, besides looking at the synoptic situation and vertical profiles. This paper provides an in-depth analysis of those events and provides extra insights of moisture intrusions to this specific region of the Arctic. I found the analysis well done and results well described. In my opinion, the paper could improve in readability by reducing the length of the abstract and summary/conclusions, and by stating more clearly the research questions and uniqueness of the study. By doing so the paper will be easier to read and the main message will appear more clearly. I explain those comments below, together with some minor comments on the paper.

*The authors thank the referee for taking the time to carefully review the manuscript. We believe the manuscript will benefit from these revisions. Below we addressed all the questions raised by the referee. Comments from the referee are in* **black** *and the responses from the authors are in* **blue***.*

**Abstract**

Is it needed to mention the dates of the field campaign in sentence 17, for me it would be enough to indicate the dates of the events which fall within the field campaign.

*The complete information with the dates of the ACLOUD campaign was removed from the abstract. We included information about the year of the events since it was not included in the previous version of the manuscript. The updated version of these sentences is shown below:*

*"During the two concerted intensive measurement campaigns, Arctic CLoud Observations Using airborne measurements during polar Day (ACLOUD) and the Physical feedbacks of Arctic planetary boundary layer, Sea ice, Cloud and AerosoL (PASCAL), which took place at and near Svalbard, three high water vapour transport events were identified as ARs, based on two tracking algorithms: on 30 May, 6 and 9 June 2017."*

Can you indicate the research question/objective more clearly in the abstract? And as well in the introduction?

The authors thank the suggestion from the referee. The research question/objectives of the manuscript were explained with more detail by including the following sentence to the Abstract:

*"The objective of this manuscript was to build knowledge from detailed AR case studies, with the purpose to perform long-term analysis. Thus, we adapted a regional AR detection algorithm to the Arctic and analysed how well does it identify ARs; used different datasets (observational, reanalyses and model) and identified the most suitable dataset; and analysed the evolution of the ARs and their impacts in terms of precipitation."*

The objectives of the manuscript were already included in the Introduction, in the last paragraph of this section:

*"Further, we apply these various observational and modelling products for investigating the ARs development and evolution, their role in the poleward moisture transport (reaching and affecting Svalbard and Greenland), and associated precipitation characteristics. Another purpose of this study is to adapt the AR tracking algorithm by Gorodetskaya et al. (2020), developed originally for Antarctica, to the Arctic region, to evaluate how well does it identify ARs and to identify the most suitable reanalysis dataset to analyse this type of events. Building on this detailed case studies analysis, it will be possible to extend this work to longer time periods from the recent past (using reanalyses) and into the future."*

Line 21-25; is all this information needed in the abstract or can you reduce the text here?

The authors tried to shorten these sentences, by removing specific information about the spatial (*"ranging from 0.25 to 1.25 degree"*) and temporal resolution (*"ranging from 1 hour to 6 hours"*). Also, the sentence *"Despite being consecutive, these events showed different synoptic evolution and precipitation characteristics"* was removed, since the following sentences in the manuscript already mention the differences in the synoptic conditions and precipitation amounts and phase.

The upgraded version of these sentences is shown below:

*"Results show that the tracking algorithms detected the events differently partly due to differences in spatial and temporal resolution, and in the criteria used in the tracking algorithms."*

Line 33: there was an increase **of values** with height. This 'of values' is not really clear. Can you improve this sentence?

The authors changed and shortened the sentence, in order to be more readable. The updated version is shown below:

*"There was an increase of wind speed with height during the first and last events, while during the second event there were no major changes in the wind speed."*

Line 35: ..during spring **and** beginning of summer ..

The authors thank the referee. The new version of the manuscript includes this correction.

**Minor comments**

Line 53: reason**s**

Thank you for the comment. In our opinion the sentence is correct since *"Gimeno et al. (2019) reason in their review (…)"* is a study performed by multiple authors, and that is why reason is in plural form.

Line 55: Here it's indicated that a typical duration of an AR is 2 to 4 days, but in your study you have two separate ARS within 3 days (6 and 9 of June). Can you comment on this? It shows up as different AR events from the figures but it would be good to emphasize that these are separate events

The information about the typical duration is not directly related with ARs. The study that mentions the 2 to 4 days duration is about moisture intrusions (Woods et al., 2013), which are not exactly the same phenomena as an AR.

In our study, we did not mention the typical duration of ARs, because it depends on the region, and there are not many studies about ARs in the Arctic. In the case of these consecutive ARs detected in a 3-day period (June 6 and 9), they were short duration events. Based on Figure 5, one can notice that the IWV and/or IVT peaks had a short duration, associated with these short events. Additionally, these two events were detected by different algorithms: the 6 June pAR was identified by Gorodetskaya et al. (2020) and the 9 June AR by Guan et al. (2018), based on different variables, namely IWV and IVT. Furthermore, both events had different pathways: the first AR moved from Western Siberia and the second from Greenland.

Line 73-75: The majority of ... --> this sentence feels unlogic here as you have been talking about ARs before but in that particular section you discuss the influence on the Arctic. I would move the sentence to line 65 or remove it from the text

The authors decided to remove the sentence. Also, the reference Zhu and Newell (1998), was removed from the manuscript since it only was referenced in this sentence.

Line 84-89: The information provided here is already very specific and would better suit in the method or discussion section

The authors understand that the information provided in lines 84-89 is too technical to be included in the Introduction (Section 1). Thus, we thank the suggestion from the referee, and moved the paragraph to the beginning of Section 3.2, where we explain in detail the algorithms used to detect atmospheric rivers. This paragraph is suitable to give a short introduction about the topic of AR tracking algorithms.

Line 90: Shields et al. (2018) study aimed to understand

The authors removed the adverb *"Furthermore"* from the beginning of the paragraph. The following paragraph was updated in the new version of the manuscript:

*"Shields et al. (2018) study aimed to understand…"*

However, the authors are not sure if this was the suggestion made by the referee.

Line 146: from 1000 hPa to 300 hPa. What are the vertical steps?

The vertical resolution of the reanalyses depends on the dataset. For ERA-Interim, ERA5, CFSv2 and JRA-55, we downloaded 20 pressure levels. From 1000 to 750 hPa the vertical steps were 25 hPa and from 700 to 300 hPa the vertical steps were 50 hPa. In the case of MERRA-2, we downloaded 21 pressure levels. From 1000 to 700 hPa the vertical steps were 25 hPa, while from 650 to 300 hPa, the vertical steps were 50 hPa.

Some information about the number of pressure levels and vertical steps was included in the updated version of the manuscript:

*"To detect the ARs, specific humidity, temperature and meridional and zonal components of the wind were acquired from 1000 hPa to 300 hPa. Except for MERRA-2, all reanalyses were downloaded from 20 pressure levels, with vertical steps of 25 hPa, from 1000 to 750 hPa, and vertical steps of 50 hPa onwards. In the case of MERRA-2, the variables were downloaded for 21 pressure levels, from 1000 to 700 hPa, with vertical steps of 25 hPa, and from 650 hPa onwards, with vertical steps of 50 hPa."*

Section 3.3 Air mass trajectories: For the air mass trajectories the NCEP dataset is used, which is not used for the rest of the analysis of detections of ARs, which is a bit inconsistent. Can you motivate your choice in the text, and did you analyse the performance of simulating the ARs in the NCEP dataset?

The authors thank the comment from the referee. Although the choice of NCEP dataset might seem a bit inconsistent, the reason why we chose this dataset is because, concerning the available datasets, this was the most suitable for our study. In the HYSPLIT online platform (https://www.ready.noaa.gov/hypub-bin/trajasrc.pl) the datasets available for the archive trajectory model are the following:

1. GDAS (1 degree, global, 2006-present)
2. GFS (0.25 degree, global, 06/2019-present)
3. GDAS (0.5 degree, global, 09/2007-06/2019)
4. NAM 12km (pressure, U.S., 05/2007-present)
5. HRRR 3km (sigma, U.S, 06/2019-present)
6. HRRRV1 3km (sigma, U.S, 06/2015-07/2019)
7. NAM 12km (hybrid sigma-pressure, U.S, 03/2010-present)
8. NARR 32km (N.A., 1979-2019)
9. EDAS 40km (U.S., 2004-present)
10. EDAS 80 km (U.S., 1997-2004)
11. NGM (N.A., 1991-1997)
12. REANALYSIS (global, 1948-present)
13. WRF 27 km (U.S., 1980-present)

Several datasets do not cover the period analysed in our study (May – June 2017), such as 2, 5, 10 and 11. At the same time, some datasets are focused on some regions, such as the U.S (4, 6, 7, 9 and 13) or North America (8). The datasets that cover the period and location of this study are 1, 3 and 12. Both datasets 1 and 3 are from the NCEP Global Data Assimilation System, and dataset 12 is from the NCEP/NCAR Reanalysis. We chose the dataset 3 (GDAS – 0.5 degree), since datasets 1 and 12 have both lower horizontal resolutions, respectively 1 degree and 2.5 degree.

The performance of NCEP to simulate ARs was not analysed in this study. All the datasets used to identify ARs were shown in the manuscript.

A short explanation about the choice of this dataset was included in the updated version of the manuscript:

*"For this study we used NCEP's Global Data Assimilation System (GDAS) model, with a horizontal resolution of 0.5 degree. Amongst the datasets available in the online platform, this was the most suitable to our study, due to its finer spatial resolution."*

Line 180: where g is the acceleration due to the gravity

The beginning of the line was changed to lowercase letter, in order to be a continuation of the equation. The sentence was updated to the following:

"where $g$ is the acceleration due to the gravity."

However, the authors are not sure if this was the suggestion made by the referee.

Line 225: here you explicitly mention ERA-Interim, but why don't you compare with ERA5?

The authors thank the comment from the referee, and we agree that ERA5 is a more suitable dataset to use in the manuscript. Besides having a higher temporal and horizontal resolution when compared with ERA-Interim, this reanalysis has been widely used over the Arctic region and it has shown better results when compared to other datasets (Graham et al., 2019). Thus, we updated Figure 1 by replacing IWV and IVT time series based on ERA-Interim by ERA5 data. The updated version of the figure is shown below.

[Figure]

References:

Graham, R. M., S. R. Hudson, and M. Maturilli, 2019: Improved Performance of ERA5 in Arctic Gateway Relative to Four Global Atmospheric Reanalyses, Geophys. Res. Lett., 46, 6138-6147, doi:10.1029/2019GL082781.

Line 297: First and second line of this section have a lot of overlap, combine them?

The authors thank the suggestion from the referee and agree that the content of the sentences is overlapping. The paragraph bellow was removed from the manuscript:

*"To understand which meteorological conditions triggered the detected events, their synoptic situation was analysed. For this purpose, we performed a detailed analysis of the synoptic conditions focusing only in the days when the pARs/ARs reached Ny-Ålesund, using ERA5 reanalysis, due to its high temporal and spatial resolution."*

An improved version of this paragraph, shown below, was included in the new version of the manuscript:

*"To understand which meteorological conditions triggered the detected events (pARs and ARs), we performed a detailed analysis of the synoptic conditions, using ERA5 reanalysis, due to its high temporal and spatial resolution."*

Line 304: ..used to study the atmospheric blocking

The adverb *"usually"* was removed from the sentence. The new version of the sentence, shown below, was included in the updated version of the manuscript:

*"The combination of these variables is used to study the atmospheric blocking…"*

However, the authors are not sure if this change comprises the suggestion proposed by the referee.

Line 354: After the landfall.. --> I believe here you mean the timing after the landfall while it could be interpret as location. This could be stated more clearly.

The following sentence was improved in order to be clearer to the readers:

*"After the time of landfall, IVT and IWV decreased sharply, which was properly represented by all datasets."*

Line 378: Are you talking about the RMSE for IWV?

The RMSE value is relative to IWV. More information about the variable being analysed was added to the new version of the manuscript, in the following sentence:

*"However, previous studies showed that IWV differences are not significant in Ny-Ålesund with an RMSE lower than 1 kg m$^{-2}$ (Nomokonova, 2020)."*

Line 520: Add interpretation to the sentence on low precip in ERA5

When comparing precipitation based on MERRA-2 and ERA5 reanalyses, one can notice that the precipitation amounts have large differences.  Furthermore, if we focus on the

surrounding of Zhelanya Cape and Dikson Island stations (figures below), these differences stand out. First, these stations are in the limits of the enhanced band of precipitation, with MERRA-2 still showing high values of precipitation over both stations, on contrary to ERA5, that already has the stations in the periphery of the precipitation band. Furthermore, even within the band of precipitation there are still substantial differences in the amounts of precipitation. Previous studies have shown that ERA5 and ERA-Interim, which is not shown here but also has similar precipitation patterns to ERA5 (see Figure 8 in the manuscript), simulate lower precipitation amounts when compared to other reanalyses over the Arctic (Boisvert et al., 2018; Barrett et al., 2020). The same studies showed that MERRA-2 reanalysis is one of the datasets with higher precipitation amounts over the entire Arctic.

[Figure]

References:

Barrett, A. P., J. C. Stroeve, and M. C. Serreze, 2020: Arctic Ocean Precipitation From Atmospheric Reanalyses and Comparisons With North Pole Drifting Station Records, J. Geophys. Res. Oceans, 125, e2019JC015415, doi: 10.1029/2019JC015415.

Boisvert, L. N, M. A. Webster, A. A. Petty, T. Markus, D. H. Bromwich, and R. I. Cullather, 2018: Intercomparison of Precipitation Estimates over the Arctic Ocean and Its Peripheral Seas from Reanalyses, J. Clim., 31, 8441-8462, doi: 10.1175/JCLI-D-18-0125.1.

Line 660: In this study we focused **on** understanding

The authors thank the correction. The new version of the manuscript includes this suggestion.

**Figures & Tables**

Figure 2: It would be nice to see the direction in which the AR is moving for extra insights in the event. You do show the orange arrows above 300, but could you lower it to get the direction for every plot? For the 6 June case you do not know the direction from the plot, which would give extra insights.

The authors thank the comment. The orange arrows (IVT) were only shown within the area of the AR detected by Guan et al. (2018), since IVT is the variable used in this algorithm. Thus, only the plots based on MERRA-2 reanalysis had these arrows, as it is explained in the figure caption (*"… orange arrows show integrated vapour transport (IVT, kg m$^{-1}$ s$^{-1}$), both based only on MERRA-2 reanalysis."*). During the June 6 event, since it was not identified as an AR by Guan et al. (2018), the arrows were not shown. In the updated version of the manuscript, we included the IVT arrows in Figure 2 during the June 6 event, within the pAR shape by Gorodetskaya et al. (2020), with the purpose to show the direction of the flow. The colours of the arrows were changed to black to be visible during the June 6 event. Also Figures S1, S2 and S3 were updated to include the IVT arrows in black. The new figures are shown below:

[Figure]

06 UTC May 30, 2017 (a)  12 UTC June 06, 2017 (b)  12 UTC June 09, 2017 (c)

ERA-Interim
ERA5
MERRA-2
HIRHAM5
IASI

Integrated water vapour (kg m$^{-2}$)

0    5    10    15    20

Guan2018    pAR Gorodetskaya2020
300 kg m$^{-1}$ s$^{-1}$    Gorodetskaya2020

[Figure]

Integrated water vapour (kg m⁻²)

[Figure]

**06 UTC June 06, 2017 (a)**     **12 UTC June 06, 2017 (b)**     **18 UTC June 06, 2017 (c)**

Integrated water vapour (kg m$^{-2}$)

0  5  10  15  20

Guan2018    pAR Gorodetskaya2020
300 kg m$^{-1}$ s$^{-1}$    Gorodetskaya2020

[Figure]

**06 UTC June 09, 2017 (a)**   **12 UTC June 09, 2017 (b)**   **18 UTC June 09, 2017 (c)**

ERA-Interim

CFSv2

MERRA-2

ERA5

JRA-55

HIRHAM5

IASI

Integrated water vapour (kg m$^{-2}$)

0   5   10   15   20

Guan2018   pAR Gorodetskaya2020

300 kg m$^{-1}$ s$^{-1}$   Gorodetskaya2020

Figure 3: Is there a better way to visualize this data? I found the figure not very attractive.

The authors thank the comment from the referee. An effort was done to make the figure more attractive. The yellow colour, previously representing CFSv2, was changed to blue to be more visible. Furthermore, we plotted each time step closer, to be easier to understand if there was an increase/decrease of the pAR/AR area during the evolution of each event.

[Figure]

Table 2: I wonder if Table 2 is needed in the main manuscript, as you only refer to it twice in the text and most information can be also found in Figure 5

The authors thank the comment from the referee. Indeed Table 2 is only referred twice in the manuscript. Since this table has important information, that is more easily read in the table that in Figure 5, we moved Table 2 to the Supplementary Material, and in the new version of the manuscript it is named Table S2.

Figure 6: end of caption --> (reference) --> still edits are needed?

The caption from Figure 6 is already completed. The *"(reference)"* in the end of the caption *"... and Figure S9 shows the differences between each reanalysis and model and the radiosondes (reference)"* is meant to explain that in Figure S9 the radiosondes are used as reference for the differences. Thus, all the results shown were calculated using the following formulation:

$$Differences = var_{model/reanalyses} - var_{radiosondes}$$

Where $var$ corresponds to the vertical profiles of specific humidity and wind speed.

Figure 9: In figure itself it should be accumulated instead of acummulated

The authors thank the correction. The colorbar title in Figure 9 was corrected. Also, the titles in Figures 8, S11, S12 and S13 were corrected.

The figures are shown below:

[Figure]

[Figure]

**Snowfall**

**06 UTC 29 May to 06 UTC 31 May (a)** **12 UTC 05 June to 12 UTC 07 June (b)** **12 UTC 08 June to 12 UTC 10 June (c)**

Accumulated snowfall (mm)

0          10          20          30

**Rainfall**

Accumulated rainfall (mm)

0          10          20          30

pAR Gorodetskaya2020
Gorodetskaya2020

[Figure]

**Accumulated total precipitation (a)**     **Accumulated rainfall (b)**     **Accumulated snowfall (c)**

**Accumulated total precipitation (a)**     **Accumulated rainfall (b)**     **Accumulated snowfall (c)**

[Figure]

ERA-Interim

CFSv2

MERRA-2

ERA5

JRA-55

HIRHAM5

Total accumulated precipitation (mm)

0   10   20   30

Accumulated rainfall (mm)

0   10   20   30

Accumulated snowfall (mm)

0   10   20   30

pAR Gorodetskaya2020 ·······   Guan2018 ■ ■ ■
Gorodetskaya2020 ■ ■ ■

[Figure]

**Accumulated total precipitation (a)**  **Accumulated rainfall (b)**  **Accumulated snowfall (c)**

---

## Author Comment (AC3)

**Comment on acp-2021-609**

**Anonymous Referee #1**

**General comments:**

This manuscript presents a detailed analysis of three atmospheric rivers affecting Svalbard during May – June 2017. A number of observational, reanalysis, and satellite datasets are used to analyze the events and the consistency between these data sources is assessed. A strength of this paper is its detailed comparison of two different AR detection algorithms, including an assessment of "potential AR" events that are hydroclimatically significant but may miss detection by the geometric criteria of certain AR algorithms. The potential impact of these events in the Arctic is a topic that has not been well explored in the literature. The detailed evaluation of several different reanalysis datasets and the HIRHAM model is also a strong contribution to the field, particularly the assessment of vertical profiles of wind and humidity. The paper is very well referenced with the appropriate literature. I have a few minor suggestions that I feel would further strengthen this work, including some quantitative validation of the various datasets and further discussion of the HIRHAM model deficiencies. Pending these minor revisions and the correction of some grammatical items as detailed below, I feel this paper will be suitable for publication in ACP.

The authors thank the referee for taking the time to carefully review the manuscript. We believe the manuscript will benefit from these revisions. Below we addressed all the questions raised by the referee. Comments from the referee are in **black** and the responses from the authors are in **blue**.

**Specific comments**

This paper has a lot of nice qualitative discussion of the differences observed among the various reanalysis and model datasets and their comparison to observations. However, quantitative assessment of these datasets is lacking. I think that providing and discussing some simple summary statistics (e.g. RMSE, bias) comparing the most important parameters that are available from both the model/reanalysis datasets and observations (IWV, IVT, wind speed) would strengthen the paper and increase its value to other researchers.

The authors thank the comments from the referee. Although in Figure S9 we already show the differences of the vertical profiles of specific humidity and wind speed based on reanalyses and model compared to the radiosondes, we agree that the manuscript would improve with the inclusion of a more quantitative discussion. For that reason, we included the suggestion from the referee and added a table with a statistical analysis of IWV and IVT for the three events identified in this study for the reanalyses, model and observations. For the vertical profiles of specific humidity and wind speed, the same

procedure was performed, but 6 plots were included in one new figure, one per event for both variables, to show the vertical profiles of BIAS and RMSE. For both analyses, we used the radiosondes as the reference, since for the vertical profiles these are the only observational dataset available.

Below we show a new table with RMSE and BIAS for IWV and IVT, to be included in the Supplementary Material of the revised version of the manuscript and referred to in Section 4.3.1 (Variability of IWV and IVT). We also show the new figure with the same statistical parameters for the vertical profiles of specific humidity and wind speed, to be included in Section 4.3.2 (Variability of vertical profiles of humidity and wind speed). Thus, the previous Figure S9 was replaced with this new figure.

**Table S1.** Integrated water vapour (IWV, kg m$^{-2}$) and integrated vapour transport (IVT, kg m$^{-1}$ s$^{-1}$) bias and RMSE during the 24 hours before and after the IWV peak at Ny-Ålesund (48 hours period), for the reanalyses (ERA-Interim, ERA5, MERRA-2, CFSv2, JRA-55), HIRHAM5 model and observations (HATPRO, GNSS and IASI), using the radiosondes as a reference (6 hours temporal resolution).

| | Variable | 30 May | | 6 June | | 9 June | |
|---|---|---|---|---|---|---|---|
| | | RMSE | Bias | RMSE | Bias | RMSE | Bias |
| Era-Interim | IWV | 1.1 | 0.2 | 1.1 | 0.2 | 0.6 | -0.5 |
| | IVT | 28.8 | 2.2 | 8.9 | 2.4 | 7.2 | -3.4 |
| ERA5 | IWV | 1.1 | -0.3 | 0.8 | -0.1 | 0.3 | -0.2 |
| | IVT | 23.2 | -1.1 | 3.5 | 0 | 3.9 | 0.2 |
| MERRA-2 | IWV | 0.9 | 0.2 | 1.2 | 0 | 1.3 | -0.6 |
| | IVT | 24.2 | 3.9 | 18.5 | 9.3 | 12.2 | -9.2 |
| CFSv2 | IWV | 0.9 | -0.3 | 1.5 | 0.1 | 0.8 | 0 |
| | IVT | 19.5 | -8.6 | 12.6 | 3.7 | 5.2 | -0.6 |
| JRA-55 | IWV | 1.5 | -1.1 | 1.4 | 0.6 | 1.2 | -0.9 |
| | IVT | 29.9 | -10.5 | 17.3 | 12.9 | 13.6 | -10.1 |
| HIRHAM5 | IWV | 5.2 | -0.2 | 4.9 | -0.2 | 11.9 | -3.5 |
| | IVT | 140.6 | 19.6 | 87.1 | 19.2 | 105.6 | -28.8 |
| HATPRO | IWV | 0.4 | 0.3 | 1.3 | 0.3 | 0.9 | 0.6 |
| GNSS | IWV | 0.8 | -0.6 | 1.5 | -1.1 | 0.9 | -0.8 |
| IASI | IWV | 1.4 | 0.6 | 6.3 | -4.0 | 5.3 | -4.1 |

[Figure]

**Figure S9.** Vertical profiles of specific humidity (g kg⁻¹, pink/orange colours) and wind speed (m s⁻¹, blue/green colours) of bias [first row, **(a)**] and RMSE [second row, **(b)**] at Ny-Ålesund based on reanalyses (ERA-Interim, ERA5, CFSv2, JRA-55, MERRA-2, dashed lines) and HIRHAM5 model (dotted lines) compared to the radiosondes (reference, 6 hours temporal resolution), during 48 hours periods (24 hours before and after the event reached Ny-Ålesund) for the 30 May event (first column), 6 June event (second column) and 9 June event (third column).

A notable feature of the results is the poor performance of the HIRHAM model compared to the reanalysis datasets, including the ERA-Interim dataset that it was forced with. This result is somewhat surprising to me since this model has high spatial and vertical resolution and has been extensively used in Arctic climate studies. I think the paper would benefit from some additional discussion of why the HIRHAM model performed so poorly. Have problems with the HIRHAM wind and humidity fields been

documented in other studies? Do the authors think this may be an issue with how HIRHAM was implemented in this particular case or a more general problem with the model?

Overall, HIRHAM5 reproduces well the IWV spatial patterns of the three AR events (Fig. 2). But the model shows clear detailed deficits and a particularly poor skill during the third event during 8-10 June 2017. For this event the model shows a spatial shift of the elevated moisture (Fig. 2c). Accordingly, the specific humidity profile (Figs. 6c and 7c), the IWV integrated during the event (Table 2), and the IWV temporal evolution (Fig. 5c) show clear deficits compared to the radiosonde observations.

As we discuss in detail in Section 4.3.1, minor differences in the location, i.e. shifts of low/high pressure systems, can induce large changes in IVT at a certain grid point. This is definitive an aspect to consider when comparing the station-nearest model grid point with the actual station observations. In addition, Ny-Ålesund is located near a fjord and characterized by complex topography, which is demanding for the HIRHAM5 model (with a relative coarse ~25 km resolution) to reproduce. Bresson et al. (2021) showed that ERA5 (with a comparable resolution of ~30 km) has a worse skill for the Ny-Ålesund station comparison than for a comparison of another station, namely Shojna, which is located in a flat in-land region. They showed the added value of higher resolution (3-6 km) simulations for the Ny-Ålesund comparison.

To compare the HIRHAM5 simulations with observations during specific events such as ARs, a prerequisite is that the model reproduces the synoptic conditions realistically. For this, simulations with nudging have been performed to keep the model close to the synoptic evolution given by the ERA-Interim forcing. This also implies that errors in the background thermodynamic state from the reanalysis may be emergent within the domain of the RCM as shown by Sedlar et al. (2020). Compared to the "Arctic Clouds in Summer Experiment" (ACSE) observations (July-October 2014), they showed that the vertical temperature and specific humidity profile statistics (mean bias and root-mean-square errors; their Fig. 3) for the RCMs (HIRHAM5 included) were similar in sign and shape to the bias profiles of the reanalysis products that were used to nudge and initialize the simulations. However, the fact that the RMSE was systematically larger in all RCMs indicates that the effect of the nudging does not inhibit the RCMs to develop their own thermodynamic state. Thus, our result that HIRHAM5 shows other (and for certain aspects even larger) biases than the forcing ERA-Interim is in agreement with this recent study.

Sedlar et al. (2020) and Inoue et al. (2021) showed that the largest temperature and specific humidity errors of HIRHAM5 (and other models) are found across the mid-troposphere to lower troposphere (their Fig. 3). The specific humidity RMSE peaks at 850-925 hPa, which is the level of moisture intrusion (our Fig. 6 and Fig. 7). Sedlar et al. (2020) also showed that the biases vary temporally (time-pressure cross-section of specific humidity differences in their Fig. S7), and that the bias was largest in connection with an occurring warm-moist air mass intrusion event. Indeed, HIRHAM5 did not always fully capture the 1-hour instantaneous IWV observations (their Fig. 2d) and showed large IWV biases for events of strong IWV (their Fig. S3). Thus, our result for the specific

three warm-moist air mass intrusion, AR events, supports and substantiates their finding that HIRHAM5 shows significant biases in reproducing such events.

Furthermore, Klaus et al. (2016, 2018) simulated monthly mean profiles of temperature from HIRHAM5 nudged runs, and showed significant temperature biases below 700 hPa (in the order of 1.5-3.5 K), especially during spring. It is reasonable to assume that associated significant specific humidity biases exist in these runs, but those were not analyzed in these studies.

Prescribed surface characteristics (SST, sea ice) and inadequate sub-grid scale model parameterizations (particularly of cloud formation, precipitation processes, and turbulence) play a role for the limited model performance.

Additional discussion about the performance of HIRHAM5 model and references to previous studies that evaluate the differences in specific humidity and IWV based on HIRHAM5 and other reanalyses/models were also included in the new version of the manuscript.

In Section 4.3.1 (Variability of IWV and IVT) the following sentence was included near the description of the temporal evolution of IWV during the 9 June event, including a reference to a previous study that mentions IWV biases in HIRHAM5 model:

*"A previous study by Sedlar et al., (2020) showed large IWV biases based on HIRHAM5 model for events of strong IWV."*

In the last paragraph of Section 4.3.2 (Variability of vertical profiles of humidity and wind) we included some results from previous studies that mention differences in the vertical profiles of specific humidity based on HIRHAM5. The following text was included in the updated version of the manuscript:

*"Previous studies by Inoue et al., (2021) and Sedlar et al., (2020) have shown that the largest specific humidity errors in HIRHAM5 occur across the mid-troposphere to lower troposphere, with the RMSE peak at around 850-925 hPa, which is in agreement with our results. Furthermore, Sedlar et al., (2020) showed that the bias of the vertical profiles of specific humidity vary temporally, and the largest values were found during events of warm-moist air intrusions."*

In Section 5 (Summary and conclusions) some sentences were included about differences in HIRHAM5 IWV and vertical profiles of specific humidity:

*"A previous study by Sedlar et al., (2020) referred to HIRHAM5 IWV biases, mainly during events of warm-moist air intrusions."*

*"An earlier study by Sedlar et al., (2020) found large errors in the HIRHAM5 vertical profiles of specific humidity."*

References:

Bresson, H., A. Rinke, M. Mech, D. Reinert, V. Schemann, K. Ebell, M. Maturilli, C. Viceto, I. Gorodetskaya, and S. Crewell, 2021: Case study of a moisture intrusion over the Arctic with the ICON model: resolution dependence of its representation, Atm. Chem. Phys. Discuss., doi:10.5194/acp-2021-501, in review.

Inoue, J., K. Sato, A. Rinke, J. J. Cassano, X. Fettweis, G. Heinemann, H. Matthes, A. Orr, T. Philips, M. Seefeldt, A. Solomon, and S. Webster, 2021: Clouds and radiation processes in regional climate models evaluated using observations over the ice-free Arctic Ocean, J. Geophys. Res. Atmos., 126, e2020JD033904, doi:10.1029/2020JD033904.

Klaus, D., K. Dethloff, W. Dorn, A. Rinke, and D. L. Wu, 2016: New insight of Arctic cloud parameterization from regional climate model simulations, satellite-based and drifting station data, Geophys. Res. Lett., 43, 5450-5459, doi:10.1002/2015GL067530.

Klaus, D., P. Wyszyński, K. Dethloff, R. Przybylak, and A. Rinke, 2018: Evaluation of 20CR reanalysis data and model results based on historical (1930-1940) observations from Franz Josef Land. Polish Polar Res., 39(2), 225–254, doi: 10.24425/118747.

Sedlar, J., M. Tjernström, A. Rinke, A. Orr, J. Cassano, X. Fettweis, G. Heinemann, M. Seefeldt, A. Solomon, H. Matthes, T. Phillips, and S. Webster, 2020: Confronting Arctic troposphere, clouds and surface energy budget representations in regional climate models with observations, J. Geophys. Res. Atmos., 125, e2019JD031783, doi:10.1029/2019JD031783.

L31: Does "the model" here refer to a deficiency in the performance of the HIRHAM model specifically, or to all the reanalysis products?

In the sentence we refer to a deficiency of the HIRHAM5 model. After analysing the vertical profiles of specific humidity (Figures S8 and S9), one can conclude that the largest differences between observations and reanalyses/model are noticed in HIRHAM5. These differences are verified in different pressure levels and are accentuated in the third event (9 June).

To clarify the readers, a reference about HIRHAM5 model was added to the sentence.

L207: The purpose and application of the successively increasing IVT percentile thresholds in the Guan algorithm are not clear from this description. How is the AR area ultimately determined from this procedure? Some additional description here would be useful.

In this study we used a refined version of Guan and Waliser (2015) (V1.0). The second version of this algorithm [V2.0, Guan et al. (2018)], has a similar methodology to V1.0. However, instead of using the fixed IVT percentile (85th), it is based on the application

of successively increasing IVT percentile thresholds (from 85th to 95th percentile, by 2.5th percentile). After the application of this threshold, the remaining methodology from V1.0 is applied.

To clarify the use of the variable threshold we changed the following sentence:

*"In this study, we used a refined version of this tracking algorithm, described in Guan et al. (2018) (V2.0), which instead of applying a fixed IVT threshold (85th percentile), it includes the application of successively increasing IVT percentile thresholds (from 85th to 95th percentile, by 2.5th percentile)."*

L281-282: What makes the Guan algorithm less restrictive compared to the polar-specific algorithms? Could the differences in algorithms have to do with using IWV (Gorodetskaya) versus IVT (Guan) for AR detection?

In the case of this event, this might be due to the lower values of IWV, which compromise the identification of the events as an AR by Gorodetskaya2020, concurrently with high values of IVT in coastal Greenland (not shown) which allowed the identification of the event as an AR by Guan2018. However, the use of a different type of threshold to identify ARs might play an important role in the restrictiveness of the algorithms. In the case of Guan2018, which is based on an absolute threshold (based on percentiles), it can be less restrictive in the Polar regions than Gorodetskaya2020, which is based on a zonal mean threshold of saturated IWV, that seems to be more suitable to identify ARs in Polar regions.

This explanation was included in the manuscript.

**Technical corrections**

L11: Specify that *atmospheric* moisture content has increased.
L19: Define what AWIPEV is an abbreviation for.
L43: "On contrary" --> "On the contrary"
L71: Specifically, ARs have been shown to influence the mass budget of *ice sheets* in the Arctic and Antarctic.
L84: "on" --> "of"
L90: Remove the word "study" from this sentence.
L92: "point" --> "pointed"
L96: "resultant of" --> "resulting from"
L111: "does it identify" --> "it identifies"
L117: "suit" --> "suite"
L147: "exception" --> "the exception"
L156: "cyclones" --> "cyclone"
L222: "on" --> "of"
L227: "is" --> "are"
L305 and elsewhere: "associated to" should be changed to "associated with".

L307: "phenomena" --> "phenomenon"
L356: "previous" --> "prior"
L394: "this type" --> "these types"
L420: Remove the word "the" before 850 hPa.
L584: "big" --> "large"
L586: "bigger" --> "larger"
L506: The word "foehn" does not need to be capitalized.

The authors thank for the corrections from the referee. All these changes were accommodated in the new version of the manuscript.

L44: Does this refer to an increase in the severity of winter weather events or in overall winter seasons?

Here we meant to refer to an increase of the occurrence of severe weather during the winter. We updated the following text in the manuscript to avoid misinterpretations:

*"On the contrary, some studies point to an increase of the probability of occurrence of severe weather during winter in the mid-latitudes (e.g. central Eurasia (Mori et al., 2019) and eastern United States (Cohen et al., 2018)), due to the Arctic warming."*

L301: I think this sentence describes figure 4 rather than figure 3.

The authors thank the correction from the referee. The number of the figure was corrected in the manuscript.

L412: I think a different word choice than "approximation" should be used here. Is this referring to the "approach" or "arrival" of the AR to Svalbard?

In the sentence we refer to the approach of the AR to Ny-Ålesund. The word was changed in the updated version of the manuscript.